

# Blind test comparison on the wake behind a yawed wind turbine

Franz  Mühle[1], Jannik Schottler[2], Jan Bartl[3], Romain Futrzynski[4], Steve Evans[4], Luca Bernini[5],
Paolo Schito[5], Martín Draper[6], Andrés Guggeri[6], Elektra Kleusberg[7], Dan S. Henningson[7],
Michael Hölling[2], Joachim Peinke[2,8], Muyiwa S. Adaramola[1], and Lars Sætran[3]

[1]Faculty of Environmental Sciences and Natural Resource Management, Norwegian University of Life Sciences, Ås, Norway
[2]ForWind - Center for Wind Energy, Institute of Physics, University of Oldenburg, Oldenburg, Germany
[3]Department of Energy and Process Engineering, Norwegian University of Science and Technology, Trondheim, Norway
[4]Siemens PLM Software, London, United Kingdom
[5]Department of Mechanical Engineering, Politecnico di Milano, Milan, Italy
[6]Facultad de Ingeniería, Universidad de la República, Montevideo, Uruguay
[7]Linné FLOW Centre and Swedish e-Science Research Centre (SeRC), Department of Mechanics, KTH Royal Institute of
Technology, Stockholm, Sweden
[8]Fraunhofer IWES, Oldenburg, Germany

*Correspondence to:* Franz Mühle (franz.muhle@nmbu.no)

**Abstract.** This article summarizes the results of a fifth Blind test workshop, which was held in Visby, Sweden, in May 2017. This study compares the numerical predictions of the wake flow behind a model wind turbine operated in yaw to experimental wind tunnel results. Prior to the work shop, research groups were invited to predict the turbines' performances and wake flow properties using computational fluid dynamics (CFD) methods. For this purpose, the power, thrust and yaw moments for a 30° yawed model turbine as well as the wake's mean and turbulent streamwise and vertical flow components were measured in the wind tunnel at the Norwegian University of Science and Technology (NTNU). In order to increase the complexity, a non-yawed downstream turbine was added in a second test case, while a third test case challenged the modelers with a new rotor and turbine geometry.

Four participants submitted predictions using different flow solvers, three of which were based on Large Eddy Simulations (LES) while another one used an Improved Delayed Detached Eddy Simulation (IDDES) model. The performance of a single yawed turbine was fairly well predicted by all simulations, both in the first and third test case. The scatter in the downstream turbine's performance predictions in the second test case, however, was found to be significantly larger. The complex asymmetric shape of the mean streamwise and vertical velocity was generally well predicted by all the simulations for all test cases. The largest improvement with respect to previous Blind tests is the good prediction of the levels of turbulent kinetic energy in the wake, even for the complex case of yaw misalignment. These very promising results confirm the mature development stage of LES/DES simulations for wind turbine wake modeling, while competitive advantages might be obtained by faster computational methods.



## 1 Introduction

Wind turbine wake interaction has become a major topic in wind energy research during the last decades. The power drop between the first and second turbine can be up to 35% in an offshore installation, when the turbines are aligned with the wind direction, while the averaged losses due to wake interactions are estimated to range between 10 - 20% (Barthelmie et al., 2009).

Furthermore, wind turbine wakes show increased levels of turbulent kinetic energy, which potentially affects fatigue loads of downstream turbines. Consequently, the prediction of the wake's mean and turbulent characteristics is highly important in the wind farm planning process in order to optimize farm layout and control. For this purpose, the development of simple analytical wake models started already 40 years ago and is still ongoing. However, these models give only predictions of the mean velocity deficit (Polster et al., 2017). For a more accurate simulation of the wake flow, advanced CFD tools based on Navier-Stokes

solvers are used. It is necessary to validate these numerical tools against experimental data sets to determine their accuracy. Therefore, a series of Blind tests providing detailed flow measurement data was initiated at NTNU in 2011. In the first Blind test the performance of a single turbine as well as the mean streamwise velocity and turbulent kinetic energy in the wake for distances up to $5D$ behind the turbine were compared, $D$ being the rotor diameter. Eight different research groups participated in the workshop, contributing various types of simulations ranging from Reynolds-Averaged-Navier-Stokes (RANS) simulations

to LES. The performance predictions showed a considerable spread around the experimental results while the prediction of wake turbulence was scattered by several orders of magnitude, as summarized by Krogstad and Eriksen (2013). For the next Blind test the complexity was increased by adding a second turbine operating in the wake of the first turbine. Modelers were asked to simulate the performance of both turbines and the wake formed behind the downstream turbine. For this Blind test nine predictions were submitted by eight organizations. The results reported by Pierella et al. (2014) still showed a large spread in

performance and also the predictions of the wake properties varied significantly. To further investigate the difference between experimental results and numerical simulations a third Blind test was realized, in which the complexity was again increased by applying a lateral offset of half a rotor diameter to the same turbine array. While the performance was predicted fairly well, the simulations of the asymmetric wake showed large uncertainties in predicting turbulence (Krogstad et al., 2015). The focus of the fourth Blind test was the influence of different inflow conditions. Therefore, the wake behind a single turbine was investigated

at three different downstream distances for a low-turbulent, a high-turbulent and a turbulent shear inflow. Furthermore the modelers were asked to predict the performance of an aligned turbine array. This Blind test attracted five groups, who all managed to predict the performance of the upstream turbine fairly well. Nevertheless, the scatter in the downstream turbine's performance was still significant. The mean wake properties were generally predicted well, while the turbulence predictions still showed a large spread, as shown by Bartl and Sætran (2017).

During the last years CFD models were constantly improved, both by increasing their accuracy and by reducing computational costs. In order to give the model developers the possibility to test their CFD models in a complex wake flow, a fifth Blind test was initiated, challenging the modelers with the dynamic flow situation of a yawed wind turbine. The wakes behind two different turbines and two inline turbines were investigated. Yaw misalignment is currently a widely discussed topic in wind energy research. Intentional yaw misalignment of an upstream turbine in a wind farm is deemed to have a large potential for



increasing the farm's efficiency (Fleming et al., 2014). A first comparison of CFD results to experimental data on yawed wind turbines was part of the so called Mexnext project (Schepers et al., 2014), in which blade loads and wake data were measured on a model wind turbine of $D = 4.5$ m operated in yaw. Even though the analysis investigated numerical flow predictions of a yawed rotor, there is need for a deeper investigation of wake properties behind yawed wind turbines. By increasing the

complexity with respect to previous Blind tests, the wake behind a yawed wind turbine is considered to be a challenging task for simulations.

## 2 Experimental setup

### 2.1 Model wind turbines

In this Blind test experiment three different turbine geometries were used. For the purpose of yaw experiments, a new turbine

test rig was constructed at NTNU, which is called Laterally Angled Rotating System 1 (LARS1). It features a shorter nacelle and slimmer tower compared to the turbines used in previous Blind tests in order to minimize the effects on the wake, as shown in Fig. 1a. A detailed description and technical drawings of all turbines are presented in the invitation document to the Blind test (Sætran et al., 2018). The 3-bladed rotor is milled from aluminum and is based on the NREL S826 airfoil. It has a diameter of $D_{LARS1} = 0.984$ m and is identical to the rotor used in previous Blind tests, a detailed description of the rotor can be found

in Krogstad and Lund (2012). At its design tip speed ratio $\lambda = 6$ and $u_{ref} = 10.0$ m/s the turbine experiences a chord based Reynolds number at the blade tips of around $Re_{tip,NTNU} = 1.1 \cdot 10^5$.

NTNU's model wind turbine called T2 was already used in previous Blind test experiments. The sketch in Fig. 1b shows that T2 has exactly the same rotor as LARS1, while the nacelle and tower structures are significantly bigger and of different shape. The turbine is used as a non-yawed downstream turbine in the investigation of an aligned turbine array.

The third turbine used in this Blind test is the model wind turbine designed by ForWind at the University of Oldenburg. For the experiments in the NTNU wind tunnel the turbine's hub height was increased with four cylindrical rods, in order to be operated at a height, comparable to the NTNU turbines. The turbine has a smaller rotor diameter of $D_{ForWind} = 0.580$ m and is sketched in Fig. 1c. The rotor is based on the SD7003 airfoil and is manufactured using a synthetic compound. A detailed description can be found in Schottler et al. (2016). It has the same design tip speed ratio $\lambda = 6$ as the NTNU turbines. For safety

reasons, it was operated at a lower inflow velocity of $u_{ref} = 7.5$ m/s, which results in a chord based Reynolds number at the tips of around $Re_{tip,ForWind} = 6.4 \cdot 10^4$.

The NTNU and ForWind rotors are based on two different airfoils. The NREL S826 airfoil, which is used from root to tip for the NTNU rotor is originally designed for the application in the tip region of full scale wind turbines, a detailed description can be found in Somers (2005). It is designed for Reynolds numbers of $Re \approx 1.0 \cdot 10^6$, which is around one order of magnitude

higher as the Reynolds number at the rotor tip in the experiments. Nevertheless, experimental data sets for airfoil performance were measured for lower Reynolds numbers at Denmark's Technical University (DTU) (Sarmast and Mikkelsen, 2012) and NTNU (Bartl et al., 2018b). In Fig. 2 the airfoil polars from the DTU experiments at $Re = 1.0 \cdot 10^5$ are compared to a standard set of lift and drag coefficients calculated for $Re = 1.0 \cdot 10^5$ in XFoil, which was provided in the invitation document (Sætran

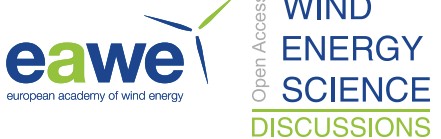

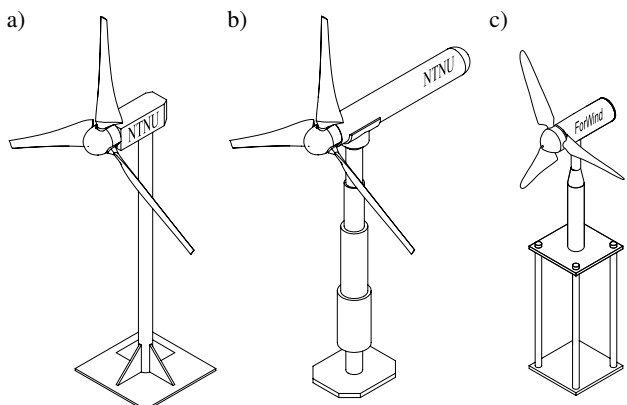

**Figure 1.** Sketches of the model wind turbines, (a) NTNU turbine LARS1, (b) NTNU turbine T2, (c) ForWind turbine.

et al., 2018). It can be seen that the drag coefficient $C_D$ is slightly different and lift coefficient $C_L$ is diverging significantly from an angle of attack $\alpha$ of approximately 4°between the experimental and XFoil data. This difference is very distinct for high angles of attack that may occur close to stall.

The ForWind rotor is based on the SD7003 airfoil that is defined in detail in Selig et al. (1995). It is specifically designed

for low Reynolds numbers and is thus well suited for wind tunnel experiments. In Selig et al. (1995) two experimental data sets for $Re = 6.4 \cdot 10^4$ and $Re = 1.02 \cdot 10^5$ are presented. They are in good agreement with XFoil data sets for $Re = 5.0 \cdot 10^4$ and $Re = 1.0 \cdot 10^5$ which were provided to the participants.

## 2.2    Wind tunnel and inflow condition

All the experimental data were measured in the closed-loop wind tunnel at the Department of Energy and Process Engineering

at NTNU in Trondheim. The wind tunnel has a test section length of 11.5 m, a width of 2.7 m and a height of 1.8 m. The reference coordinate system is pictured in Fig. 3 and a detailed description can be found in Sætran et al. (2018).

For all test cases a non-uniform shear flow was generated by a grid at the inlet of the test section. The grid is built from wooden bars with a cross section of 0.047 m x 0.047 m. In the horizontal direction the bars are evenly distributed with a distance of 0.24 m between the edges of the bars. In the vertical direction the mesh size increases with increasing height from a clearance

of 0.016 m close to the floor to an opening of 0.30 m underneath the roof. The grid has a total solidity of about 34% in the wind tunnel cross section. The shear profile can be described by the power law,

$$\frac{\bar{u}}{u_{ref}} = \left( \frac{y}{y_{ref}} \right)^{\alpha} . \tag{1}$$

The power law describes the wind speed $\bar{u}$ as a function of the height $y$ provided that the reference wind speed $u_{ref}$ is known at a reference height $y_{ref}$. The strength of the shear is described by the power law coefficient $\alpha$. The shear grid used in the

experiments was designed to obtain an exponent of $\alpha$ 0.11.

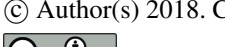


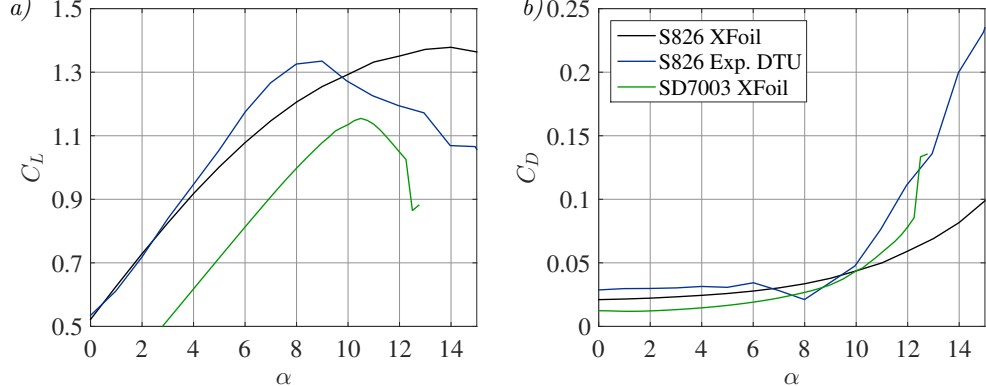

**Figure 2.** (a) lift coefficient and (b) drag coefficient for $Re = 1.0 \cdot 10^5$ for NREL S826 from XFoil, NREL S826 from experiments DTU and SD7003 from XFoil.

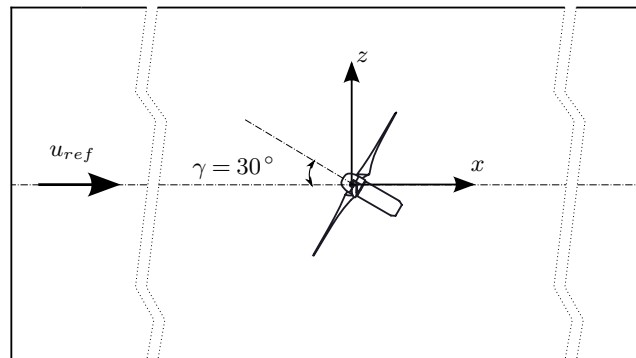

**Figure 3.** Reference coordinate system in the wind tunnel and definition of positive yaw angle $\gamma$, top view.

As the velocities of the shear profile vary in height and are non-uniform over the rotor area, the reference wind speed $u_{ref}$ is defined at the turbine hub height as shown in Fig. 4a. Furthermore, the velocity profile approximated by Eq. (1) matches well with the measured velocities, having a maximum deviation of $\pm 1.0\%$. Fig. 4b shows the normalized vertical velocity component of the inflow for the NTNU turbine. It can be seen that the vertical flow component $v$ is negative, which creates a

5    slight down flow in the wind tunnel. The deviations in $v$ from zero were not known at the time the Blind test invitation was sent out, in which a zero velocity component for $v$ was assumed. In order to take this into account, in the comparison, $v$ at the inlet is subtracted from the vertical velocity component that is measured in the wake at the same $y$-position.

The turbulence intensity ($TI$) of the inflow is shown in Fig. 4c. As expected, the turbulence decays with increasing downstream distance. At the position of the NTNU turbine the turbulence intensity is measured to be $TI = 10.0\%$ at hub height. The integral

10    length scales $L_{uu}$ are calculated from hot-wire measurements of the streamwise velocity fluctuation $u'$ and the dissipation rate of the turbulent kinetic energy $E$, by applying $E = {}^3\!/_2 A \frac{u'^3}{L_{uu}}$, where $A \approx 1$, taken from Krogstad and Davidson (2010). This results in $L_{uu} = 0.097$ m at the position of the NTNU turbine. The ForWind turbine was placed $5D$ ($D = D_{LARS1}$) behind the



shear grid and thus experienced a lower turbulence intensity of TI = 5.2%. The integral length scale however increased to $L_{uu} = 0.167$ m at this position. The third investigated streamwise position is $6D$ behind the NTNU turbine. At this position the turbulence has further decayed to $TI = 4.1\%$. The corresponding integral length scale at this position is $L_{uu} = 0.271$ m.

## 2.3 Test cases description

In this Blind test experiment the modelers were asked to simulate three test cases. In test case 1 the flow $3D$ and $6D$ behind the yawed turbine LARS1 and its performance, thrust force and yaw moment are investigated. The grid at the inlet is located -2D upstream of the turbine location at $x = -2D$. The inflow velocity is adjusted to $u_{ref} = 10.0$ m/s and the turbulence intensity is $TI = 10.0\%$ at the turbine's position. The turbine's hub height is in the center of the wind tunnel at $h_{hub} = 0.89$ m. LARS1 is yawed to $\gamma_{LARS1} = +30°$ and operated at its design tip speed ratio of $\lambda_{LARS1} = 6$ throughout all measurements. In test case

2 a turbine operating in the wake of a yawed upstream turbine is investigated. Therefore, the setup of test case 1 is extended with the turbine T2 located $3D$ behind the upstream turbine LARS1. In contrast to LARS1, T2 is not yawed ($\gamma_{T2} = 0°$). As the downstream turbine is impinged by a partial wake of the upstream turbine, its optimum tip speed ratio is reduced to $\lambda_{T2} = 5$, taking into account that the tip speed ratio is based on the constant reference velocity $u_{ref} = 10.0$ m/s upstream of the two-turbine array. This test case investigates to which degree a partial wake impact can deflect the wake behind a non-yawed downstream

turbine. This has recently been investigated in a LES study by Fleming et al. (2017). In test case 3, similar to test case 1, the flow $3D$ and $6D$ ($D = D_{ForWind}$) behind the ForWind turbine is investigated. The turbine is located at $x = 3D$ ($D = D_{LARS1}$), which resulted in a lower turbulence intensity of $TI = 5.2\%$ at the turbine position. The hub height is set to $h_{hub} = 0.89$ m and the inflow velocity is reduced to $u_{ref} = 7.5$ m/s. Corresponding to test case 1 the turbine is yawed for $\gamma_{ForWind} = 30°$ and is operated at its optimum tip speed ratio of $\lambda_{ForWind} = 6$. All setup parameters for test cases 1-3 are summarized in Table 1 and a detailed

description can be found in (Sætran et al., 2018).

## 3 Methods

### 3.1 Measurements techniques

The $u$- and $v$- velocity components in the wake were measured using a 2-component FiberFlow Laser Doppler Velocimetry (LDV) system from DANTEC dynamics. The LDV probe was placed inside the wind tunnel on a traverse system. For each

measurement point $5.0 \cdot 10^4$ samples were recorded. The sampling frequency was adjusted by controlling the particles in the flow, ranging from 1500 - 2000 Hz which resulted in an average sampling time of approximately 25 - 33 s.

The thrust force and yaw moments acting on the upstream and downstream turbine were measured separately using a Schencker six-component force balance, which was installed under the wind tunnel floor. The balance also served as a turning table allowing an exact adjustment of the yaw angle. For the rotor thrust only the load cell parallel to the flow was taken into

account. The yaw moment was calculated from a moment equilibrium of three measured forces in the horizontal plane (referred to the rotor center).





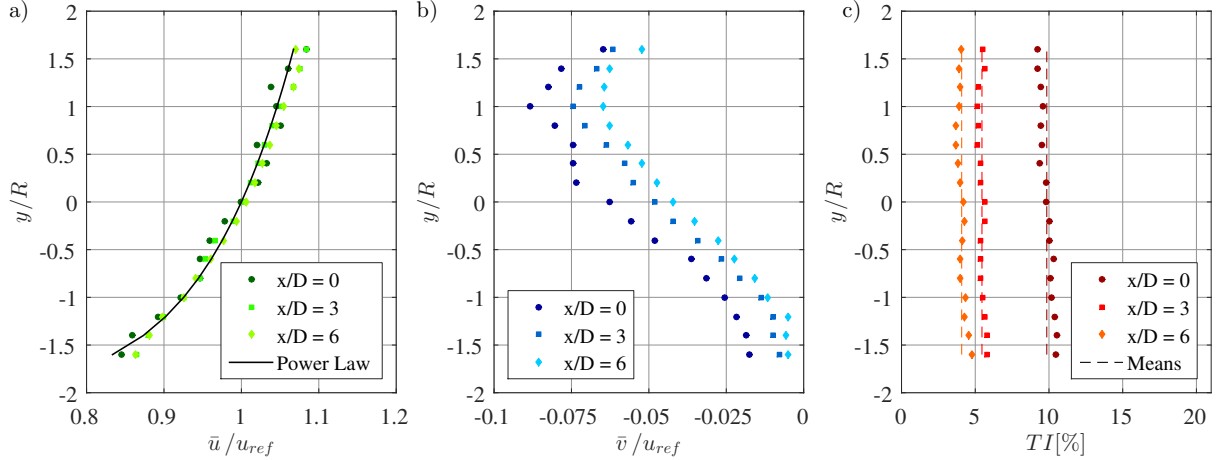

**Figure 4.** Inflow at different wind tunnel positions, in which $x/D = 0$ refers to the position of the NTNU turbine: (a) normalized streamwise velocity $u^*$, (b) normalized vertical velocity $v^*$, (c) turbulence intensity $TI[\%]$. The radius $R$ and diameter $D$ refer to the NTNU turbine.

**Table 1.** Summary of the parameters that are varied for the three investigated test cases, *up* refrs to the upstream and *down* to the downstream turbine position.

| Test case | $u_{inf}$ | $TI$ at turbine position | Upstream turbine | $\lambda_{up}$ | $\gamma_{up}$ | Downstream turbine | $\lambda_{down}$ | $\gamma_{down}$ | Wake scan locations |
|-----------|-----------|--------------------------|------------------|----------------|---------------|--------------------|------------------|-----------------|---------------------|
| TC1 | 10.0 m/s | 10.0% | LARS1 | 6.0 | 30° | - | - | - | 3D, 6D |
| TC2 | 10.0 m/s | 10.0% | LARS1 | 6.0 | 30° | T2 | 5.0 | 0° | 6D |
| TC3 | 7.5 m/s | 5.2% | ForWind | 6.0 | 30° | - | - | - | 3D, 6D |

The aerodynamic power $P$ of the NTNU rotors was measured using the test rig of turbine T2. This turbine is equipped with an optical RPM sensor and a torque transducer in the hub. Thus, the torque $T$ and the rotational speed $\omega$ of the turbine could be measured simultaneously so that $P = \omega \cdot T$.

### 3.2 Measurement uncertainties

The experimentally measured values feature several uncertainties. The statistical uncertainties of every sample of the mean velocity, power, thrust and yaw moments are calculated based on a 95% confidence level according to the procedure described in Wheeler and Ganji (2010). The uncertainty for the power measurements is calculated to be within ±3% while the force measurements' uncertainty is slightly lower (±2%). The exact values for all measured points are presented as error bars in the plots for the power coefficients $C_P$, the thrust coefficients $C_T$ and the yaw moments $M_y^*$. The uncertainties for the mean streamwise velocities $u$ in the wake are calculated to be smaller than ±1%. The uncertainties for the vertical velocity





component $v$ are slightly higher due to the correction by the inlet component. In order to determine the inaccuracy in the turbulent kinetic energy measurements, the method proposed by Benedict and Gould (1996) was applied. The uncertainties for a 95% confidence level are found to be below $\pm 2\%$ in the wake. It should be noted that the coarse measurement grid slightly influences the position of the turbulent kinetic energy (TKE) peaks.

## 3.3 Participants and computational methods

Siemens PLM software from the United Kingdom (Siemens), the Department of Mechanical Engineering of the Politecnico di Milano in Italy (POLIMI), the Facultad de Ingeniería of the Universidad de la República in Uruguay (UdelaR) and KTH Mechanics from the Royal Institute of Technology in Sweden (KTH) participated in the Blind test and submitted computational results. For clarity, only the abbreviations will be used in the following. A summary of the simulation methods and mesh properties is presented in Table 2.

### 3.3.1 Siemens PLM Software (Siemens)

Siemens, who previously participated in Blind test experiments as CD-adapco, used the finite volume code STAR-CCM+ v12.04 to mesh and solve all three test cases. Each simulation resolved the rotor, nacelle and tower structure completely, and used the hybrid method Improved Delayed Detached Eddy Simulation (IDDES), which resolves the energy-carrying eddies in the free stream and solves the boundary layer flow with RANS. The Spalart-Allmaras model was used for closure of the turbulence equations, and the fluid was considered incompressible. Convective fluxes used a MUSCL 3$^{rd}$ order scheme, while time was discretized using a 2$^{nd}$ order implicit scheme. Each set of blades and hub was contained inside a cylindrical, rotating volume which was meshed with polyhedral cells, whereas the main domain used trimmed cells, resulting in a hexahedral dominant mesh in which a small proportion of cells was trimmed near the boundaries. Due to the rotation of the cylindrical volumes, the mesh was not conformal at the interface between the two regions, and flow quantities were interpolated from one volume to another. All wall surfaces, including the wind turbine bodies and the wind tunnel walls, were covered in several layers of prismatic cells to improve the resolution of boundary layers. The resulting $y^+$ values were below 1 on the turbine bodies, and around 30 on the wind tunnel walls. The smallest cell size on the surface of the turbine bodies was 0.3 mm, typically found at the leading edge of the blades. The characteristic cell size in the rotating regions was 10 mm, which was also the cell size used in the wake of the rotors. The rest of the domain had a characteristic cell size of 20 mm. This resulted in meshes of $29 \cdot 10^6$, $35 \cdot 10^6$, and $17 \cdot 10^6$ cells for cases 1, 2, and 3 respectively. All simulations were run with a time step of $10 \cdot 10^{-4}$ s. As inflow the given analytical mean velocity profile $U_{inlet} = u_{ref} \cdot (y \text{-} y_{ref})^\alpha$ was used. Furthermore, the Synthetic Eddy Method was used to superpose time-dependent eddies with the characteristic length scale of 10 mm, and a turbulence intensity $TI = 5\%$. All cases were run for 1.6 s to establish the flow prior to sampling, and then mean values were sampled over a period of 2 to 3 s. An example using STAR-CCM+ can be found in (Mendonça et al., 2012).





**Table 2.** Overview of simulation methods and parameters. Abbreviations: Improved Delayed Detached Eddy Simulation (IDDES), Large Eddy Simulation (LES), Actuator Line (ACL), Fully Resolved (FR).

| Participant | Simulation code | Flow solver type | Rotor model | Airfoil polars | Tower, nacelle | Mesh properties | Number of cells |
|---|---|---|---|---|---|---|---|
| Siemens | Star-CCM+ | IDDES | FR | - | FR | Hexah./polyh. | $\approx 30.0 \cdot 10^6$ |
| POLIMI | ALEVM | LES | ACL | X-Foil | No | Cartesian | $\approx 4.1 \cdot 10^6$ |
| UdelaR | caffa3d | LES | ACL | X-Foil | Yes | Cartesian | $\approx 0.7 \cdot 10^6$ |
| KTH | Nek5000 | LES | ACL | Experiments | Yes | Uniform | $\approx 58.0 \cdot 10^6$ |

### 3.3.2 Politecnico di Milano (POLIMI)

POLIMI submitted a LES that was computed using the ALEVM code. It is an aerodynamic turbine simulation tool written in C++ and based on pisoFoam, which is an incompressible transient solver included in the OpenFOAM framework. The standard PISO (Pressure-Implicit with Splitting of Operators) solver was modified to include the effect of the turbine blades that are

represented using the lifting line approach. The blade lines are discretized in segments based on the intersections with the numerical mesh grid, in which an actuation point acts on each segment. Each point of the Actuator Line (ACL) acts as an isolated blade section. More information about the ACL method can be found in Sørensen and Shen (2002). The wind velocity is numerically sampled for every blade point and used to compute the relative wind speed and the angle of attack. Thereafter, the aerodynamic forces are obtained through a lookup table, in which the blades' geometrical and aerodynamic properties are

listed. In ALEVM the wind velocity is not sampled on a single point but averaged over a line, which is placed upstream of the blade point position with a distance proportional to the mesh cell dimension. The wind velocity is estimated using the mean of the velocity probed across the line. The main purpose of the relative wind speed estimation is in the angle of attack calculation. The wind velocity direction is then corrected to account for the local up wash due to the lifting line force. Based on the lifting line approach, the ALEVM code includes the turbine blade effect as an external momentum source term in the Navier-Stokes

equations solved by the PISO algorithm.

ALEVM employs the well know solution of the Regularization kernel, smearing the line forces on the multiple cells following a Gaussian distribution and thus avoiding abrupt variation of the source term strength between adjacent cells. The turbulence in the wake region is modeled using a LES, adopting the Smagorinsky sub-grid scale model. For the time discretization scheme a first order implicit approximation is used, while the divergence discretization scheme and the gradient discretization scheme

are approximated by second order. The wind tunnel walls are included as no-slip-boundaries, while also the inlet turbulence grid is geometrically modeled. The total cell count for the simulations is approximately $4.1 \cdot 10^6$. Further details about the code can be found in (Schito and Zasso, 2014).





### 3.3.3 Universidad de la República (UdelaR)

UdelaR submitted another LES using their in-house developed caffa3d code. It is an open source, finite volume code, with second order accuracy in space and time, parallelized with a Message Passing Interface (MPI), in which the domain is divided in unstructured blocks of structured grids. Complex geometries are represented by a combination of body fitted grids and the
immersed boundary method over both, Cartesian and body fitted grid blocks. The properties of the geometry and the flow are expressed as primitive variables in a Cartesian coordinate system, using a collocated arrangement. An ACL approach is used to discretize the turbine blades in the simulations. The aerodynamic forces on the blade elements are computed using the provided XFoil data. The forces then are projected onto the computational domain. In order to compute the additional source term, a Gaussian smearing function is used, taking into account one smearing factor for each direction: normal, tangential and radial to
the rotor plane. The domain, representing the wind tunnel, is uniformly divided into 192 x 72 x 48 grid cells in the streamwise, spanwise and vertical directions, resulting in a total cell count of approximately $0.7\cdot10^6$. A zero velocity gradient is imposed at the outlet, while a logarithmic law is used to compute the stress at the bottom wall and the symmetry boundary condition is used at the lateral and top boundaries. A Crank-Nicolson time scheme is used with a time step of $2.5\cdot10^{-3}$ s. The scale dependent dynamic Smagorinsky model is used to compute the subgrid scale stress, using a local averaging scheme. The inflow condition
is obtained from a precursor simulation with a similar numerical setup. More information about the application of caffa3d for wind energy simulations can be found in (Guggeri et al., 2017; Mendina et al., 2014; Usera et al., 2008).

### 3.3.4 Royal Institute of Technology (KTH)

A third LES was submitted by KTH. The spectral element code Nek5000 (Fischer et al., 2008), which was developed to solve the dimensionless, incompressible Navier–Stokes equations, was used. Each spectral element is discretized using Gauss–
Lobatto–Legendre quadrature points on which the solution is expanded using Legendre polynomials. The LES applies a spatial filtering technique to the two highest modes to remove a part of the energy in the smallest scales and redistribute it to the lower modes thus stabilizing the numerical simulation. The domain is discretized using $7.98\cdot10^4$ uniformly distributed spectral elements with 9[th] order polynomials in each element, resulting in a total cell count of approximately $58\cdot10^6$. The numerical domain size corresponds to the dimensions of the wind tunnel. In the case of the NTNU turbine this mesh size corresponds to
45 grid points along each blade, when the blades are aligned with the mesh. The wind turbine blade geometry is represented by body forces according to the ACL method with the lift and drag forces being computed using tabulated airfoil data. For the NTNU turbines the experimental airfoil data set from DTU (Sarmast and Mikkelsen, 2012) is used. It provides lift and drag coefficients over a range of Reynolds numbers. The ForWind turbine lift and drag forcing was computed using airfoil polars generated by Xfoil that were provided in the invitation. At the blade tips the Prandtl tip correction is applied. The forces
computed at each actuator line are distributed using a three-dimensional Gaussian distribution. The Gaussian width is selected to be 2.5 times the average grid spacing. The tower is also modeled using a body force approach. Both an oscillating lift component and a constant and oscillating drag component are included. The lift and drag coefficients for the mean drag and root-mean-squared lift and drag of a cylinder are taken from Summer and Fredsøe (2011). The line forces are then distributed





using the three-dimensional Gaussian approximately in the volume occupied by the tower. This setup has been previously validated against experimental data from the NTNU turbine (Kleusberg et al., 2017). In the case of the ForWind turbine only the actual tower of the support structure is included. The turbulence at the inlet is modeled using sinusoidal modes with random phase shifts and they are scaled with a von Kármán energy spectrum. It is superimposed to the desired uniform inflow condition.

The turbulence is calibrated to give a turbulence intensity at hub height of approximately $TI = 10.0\%$ at the upstream turbine LARS1 and $TI = 4.8\%$ at the downstream turbine T2. At the outlet a zero-stress boundary condition is used while the symmetry boundary condition is imposed laterally to avoid resolving the wall boundary layer. More details about the the computational setup can be found in (Kleusberg et al., 2017).

### 3.3.5   Wind turbine performance, forces and moments

The modelers were asked to predict the power coefficients $C_P$ (Eq. 2), where $P$ is the mechanical power of the turbine, $\rho$ is the air density and $A$ the rotor swept area, as well as the thrust coefficients $C_T$ (Eq. 3), where $T$ is the thrust force acting on the whole test rig, including rotor and tower, perpendicular to the rotor plane. Furthermore, the normalized yaw moments $M_y{}^*$ (Eq. 4), were required, where $M_y$ is the yaw moment that is calculated by a moment equilibrium of the horizontal forces taking the distances of the load cells according to the center of the rotor plane into account. In test case 1 the power coefficient

$C_{P,LARS1}$, the thrust coefficient $C_{T,LARS1}$ and the normalized yaw moment $M_{y,LARS1}{}^*$ are compared. For the aligned turbine array in test case 2, the predictions for the upstream turbine are similar to test case 1. However, additional predictions of $C_{P,T2}$, $C_{T,T2}$ and $M_{y,T2}{}^*$ for the downstream turbine were compared. Due to a high uncertainty in the power and thrust force measurements of the ForWind turbine, $C_{P,ForWind}$, $C_{T,ForWind}$ and $M_{y,ForWind}{}^*$ are not compared in test case 3. The performance characteristics of the NTNU turbines are listed in Table 1.

$$C_P = \frac{2P}{\rho \cdot A \cdot u_{ref}{}^3} \tag{2}$$

$$C_T = \frac{2T}{\rho \cdot A \cdot u_{ref}{}^2} \tag{3}$$

$$M_y^* = \frac{M_y}{\rho \cdot A \cdot u_{ref}{}^2 \cdot D} \tag{4}$$

### 3.3.6   Mean and turbulent wake flow

The modelers were asked to provide predictions of the velocities and turbulent kinetic energy in full wake planes in the ranges $-1.0 \le z/D \le +1.0$ and $-0.8 \le y/D \le +0.8$. The grid points are separated by $0.1D$ resulting in a grid consisting of 357 points, which is sketched in Fig. 5. The streamwise and vertical velocities $u$ and $v$ for all points are normalized by $u_{ref}$ so that $u^* = u/u_{ref}$ and $v^* = u/u_{ref}$ respectively. The same procedure is applied for the turbulent kinetic energy $k$, which is

normalized to $k^* = k/u_{ref}{}^2$. The turbulent kinetic energy in a three dimensional flow is defined as

$$k = {}^1\!/_2 \left( \overline{u'^2} + \overline{v'^2} + \overline{w'^2} \right). \tag{5}$$



However, in the experiments only the two velocity components *u* and *v* were measured. Comparing *u'* and *v'* showed that the TKE is not perfectly isotropic. Therefore, additional measurements of the third velocity component *w* for one wake scan were performed to investigate whether the fluctuations *v'* and *w'* were in the same range. The results confirmed the assumption, allowing an approximation of the turbulent kinetic energy as

$$k = \frac{1}{2}\left(\overline{u'^2} + 2\overline{v'^2}\right).$$
(6)

### 3.4 Comparative methods

Two-dimensional wake contours are difficult to compare quantitatively as they cannot be plotted in the same diagram. However, they provide valuable insight into the shape and position of the wake. Therefore, the wake shapes are in a first iteration compared qualitatively. To obtain quantitative measures of comparison, different methods to compute the wake position, the

energy content in the wake and the magnitudes of the wake parameters are applied. These are described below.

#### 3.4.1 Available power method for wake deflection and energy content

In order to quantify the wake deflection, a method approximating the available power is used, which was previously described by Schottler et al. (2017). This method is deemed to be an appropriate approach to analyze the wake deflection of a yawed wind turbine, as it takes the full wake scans into account. To find the wake center deflection an imaginary rotor is traversed

laterally in the wake while the wake center is defined as the position where the available power in the wake is the lowest. To get information about the energy content in the wake, the minimum of available power of the deflected wake is normalized by the available power found in the free stream of the experiment. With the resulting normalized minimum available power ($P_{wake}{}^*$) possible deviations in the location and magnitude of the energy content can be directly quantified.

#### 3.4.2 Statistical methods for wake properties

The predictions of $C_P$, $C_T$ and $M_y{}^*$ are directly compared to the experimental results. The deviations of the predictions from the measurements are presented as a percentage of the experimental reference value in supplementing tables.

From the statistical error measures proposed by Chang and Hanna (2004) the normalized mean square error (*NMSE*) and the correlation coefficient (*r*) are used to quantify the differences between simulations and experiments regarding $u^*$, $v^*$ and $k^*$. For this purpose, all 357 points in the *yz*-plane of the CFD predictions are compared to the corresponding measurement points.

Perfect predictions would result in *NMSE* = 0.0 and *r* = 1.0. They are calculated according to

$$NMSE = \frac{\overline{(x_e - x_s)^2}}{\overline{x_s} - \overline{x_e}},$$
(7)

$$r = \frac{\overline{(x_e - \overline{x_e}) \cdot (x_s - \overline{x_s})}}{\sigma_{x_e} \cdot \sigma_{x_s}},$$
(8)





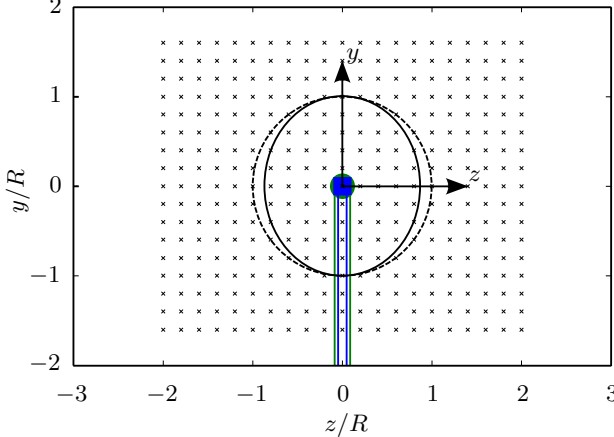

**Figure 5.** Measurement grid in the wake consisting of 357 points, the blue tower and nacelle represents the NTNU turbine LARS1, the green tower and nacelle represents the ForWind turbine, the dashed line corresponds to the projection of the rotor diameter $\gamma = 0°$, the solid line corresponds to the projection of the rotor area $\gamma = 30°$ and the outline corresponds to the cross section of the wind tunnel with the NTNU turbine installed.

where $x_e$ represents the experimentally measured values and $x_s$ are the simulated values. $\overline{x}$ indicates the average of all 357 points of the full wake scans. The standard deviation of all points of the whole wake scan is given in $\sigma_x$. *NMSE* is a measure of mean relative scatter and thus reflects both systematic and random errors (Chang and Hanna, 2004), as the difference of every data point is squared, outliers are emphasized, which is not considered to be significant as no major outliers are expected. *NMSE*

5   is used to analyze the predictions of $u^*$ and $k^*$. The method is however not suited to evaluate the discrepancy of $v^*$, because $v^*$ fluctuates around 0. Consequently the denominator of Eq. (7) also ranges around 0 which results in unrealistically high values for the *NMSE*. The correlation coefficient $r$ represents a linear relationship between the measurements and predictions. It directly compares the measured and predicted values at a certain point. The predictions of all three investigated wake properties $u^*$, $v^*$ and $k^*$ are analyzed using the coefficient $r$.

## 10  4   Results

### 4.1   Test case 1

#### 4.1.1   Power, thrust and yaw moment

The results of $C_{P,LARS1}$, $C_{T,LARS1}$ and $M_{y,LARS1}{}^*$ for test case 1, in which the turbine is operated at $\gamma = 30°$, are depicted in Fig. 6. For $\lambda = 6$ the differences between the experimental and numerical results are summarized in Table 3. Comparing the

15   values of $C_{P,LARS1}$ in Fig. 6a it can be seen that the simulation results deviate from the measurements by up to 19%. This is a larger scatter compared to the previous Blind tests eg. (Bartl and Sætran, 2017). However, it should be kept in mind that the complexity is increased by the yawed turbine operation. Siemens, who fully resolved the rotor, overpredict $C_{P,LARS1}$ by 14.2%,



which is almost in the same range as UdelaR and POLIMI who used ACL with the provided polars from XFoil and showed deviations of 18.5% and 16.8%, respectively. KTH also applied an ACL model, but used the experimentally generated data set of airfoil polars from DTU (Sarmast and Mikkelsen, 2012). Using this data results in a good agreement with the experimental data with only a slight underprediction of 2.3%.

The BEM tool Ashes (Thomassen et al., 2012) was used to analyze the blade loads. The calculations showed that the angle of attack for the yawed turbine is fluctuating during one rotation for approximately 2.0°in the outer third of the blade, causing very high angles of attack to occur on the blade. From Fig. 2 can be seen that the lift and drag coefficient from the DTU experiments and XFoil are very different for such high angles of attack. The experimental polars from DTU seem to be more accurate as the polars predicted with XFoil for such high angles of attack, what explains the better predictions of $C_P$ by the

simulations using the experimental polars.

The thrust coefficients $C_{T,LARS1}$ for the single yawed turbine LARS1 are presented in Fig. 6b and only show a small scatter of up to 7.0% around the experimental results and thus are almost all within the measurement uncertainty. Consequently, for $C_T$ predictions the experimental polars do not yield better results with respect to the polars generated by XFoil. The yaw moment $M_{y,LARS1}^*$ is presented in Fig. 6c, over a range of yaw angles from $\gamma$ = -40° to $\gamma$ = +40°. All simulations underestimate the

experimental value of $M_{y,LARS1}^*$ while the deviations ranging from about 30% to 80% are rather large. Nevertheless it should be kept in mind that the values of $M_{y,LARS1}^*$ are very small and thus small deviations result in large differences in percentage.

### 4.1.2   Wake characteristics

Fig. 7 shows a comparison of the predictions of the streamwise velocity $u^*$ at $x = 3D$, with line profiles at hub height are added to the full wake contours. The wake contours as presented in Fig. 7b show a slightly curled wake shape, which is generally well

predicted by three of the simulations. Only the wake predicted by UdelaR has a rather oval shape. As expected, the wake is not only curled, but also clearly deflected in negative $z$-direction. This is very well predicted by all the simulations. POLIMI and KTH match the deflection whereas UdelaR and Siemens slightly underestimate it. This is not consistent with the predictions of $C_T$ in which all institutions except Siemens estimate a lower $C_T$. The tower shadow is also clearly visible in all simulations. By fully resolving the rotor and turbine geometry Siemens matches the experimental results almost perfectly. UdelaR and KTH,

who both modeled tower and nacelle with a line of drag forces, simulate a fairly accurate tower shadow. Even though POLIMI did not model nacelle and tower, their results show a strong velocity deficit in the area where the tower shadow is expected. This effect is considered to be caused by the flow velocities modeled near the wind tunnel floor, whose influence is pronounced in all simulations by POLIMI. In the free stream, the shear flow can be clearly seen in the experimental results. Siemens, UdelaR and KTH apply a user defined shear function at the inlet and thus predict a smooth shear profile, while POLIMI, who

fully resolved the turbulence grid at the inlet, simulate a shear profile with a too strong shear and very low velocities close to the floor. Fig. 7a shows that POLIMI generally predicts lower velocities in the free stream, as the normalized velocity $u^*$ at hub height does not reach 1.0 in the free stream. Nevertheless, the velocities behind the rotor are represented very well, while a poor $NMSE_u$ of 0.017 and a $r_u$ of 0.878 show the discrepancy in the free stream to the measurements. All in all, it can be seen that $u^*$ is predicted well by all simulations. Siemens' results for this test case are almost perfectly in accordance with the

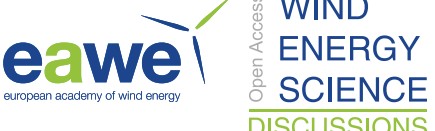

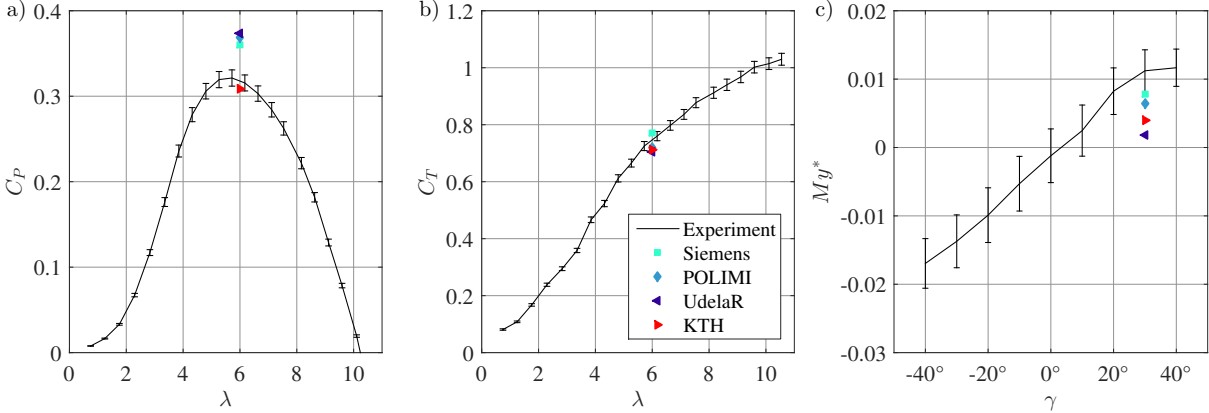

**Figure 6.** Power coefficient $C_{P,LARS1}$ (a) and thrust coefficient $C_{T,LARS1}$ (b) for the upstream turbine LARS1 operated at $\gamma_{LARS1} = 30°$ and normalized yaw moment $M_{y,LARS1}{}^*$ (c) for the upstream turbine LARS1 operated at $\lambda_{LARS1} = 6$ for $\gamma = -40°$ to $+40°$.

**Table 3.** Numerical values of power coefficient $C_P$, thrust coefficient $C_T$ and normalized yaw moment $M_y{}^*$ and deviations of predictions to measurements in percent for test cases 1 and 2.

| Institution | Upstream turbie LARS1 | | | | | | Downstream turbie T2 | | | | | |
| --- | --- | --- | --- | --- | --- | --- | --- | --- | --- | --- | --- | --- |
| | $C_{P,LARS1}$ | diff | $C_{T,LARS1}$ | diff | $M_{y,LARS1}{}^*$ | diff | $C_{P,T2}$ | diff | $C_{T,T2}$ | diff | $M_{y,T2}{}^*$ | diff |
| Experiments | 0.32 | | 0.76 | | 0.011 | | 0.19 | | 0.63 | | 0.011 | |
| Siemens | 0.36 | 14.2% | 0.77 | 1.7% | 0.008 | 30.5% | 0.21 | 10.5% | 0.56 | -10.7% | 0.022 | 101.4% |
| POLIMI | 0.37 | 16.8% | 0.72 | -5.2% | 0.006 | 42.5% | 0.27 | 43.6% | 0.60 | -4.6% | 0.016 | 43.4% |
| UdelaR | 0.37 | 18.5% | 0.71 | -7.0% | 0.002 | 84.1% | 0.28 | 48.9% | 0.56 | -10.6% | 0.005 | -50.6% |
| KTH | 0.31 | -2.3% | 0.71 | -6.1% | 0.004 | 65.1% | 0.19 | 0.0% | 0.53 | -15.3% | 0.011 | 1.5% |

experiments, which results in a very low $NMSE_u$ of 0.002 and a large $r_u$ of 0.964. Good statistical performance values are also achieved by KTH ($NMSE_u = 0.002$, $r_u = 0.957$), even though the velocity deficit in the wake center is slightly underestimated. An even clearer under prediction of the velocity deficit in the wake center can be observed for the UdelaR simulations, which result in a $NMSE_u$ of 0.005 and a $r_u$ of 0.914. These observations are confirmed by comparing the available power levels in the wake (Table 4). In case of Siemens' accurate simulations of $u^*$, $P^*_{wake}$ only deviates by -2.7% from the experiments. UdelaR underestimates the velocity deficit in the center significantly, resulting in an overprediction of $P^*_{wake}$ by 42.7%. KTH also overestimates $P^*_{wake}$ by 15.6%, which confirms the higher velocities observed in the wake center. The available power method shows a good agreement of POLIMI's simulations with the experiments, deviating only 11.2%. This is because the method takes only the area in the wake center into account and thus is not affected by the deviating velocity levels in the free stream.





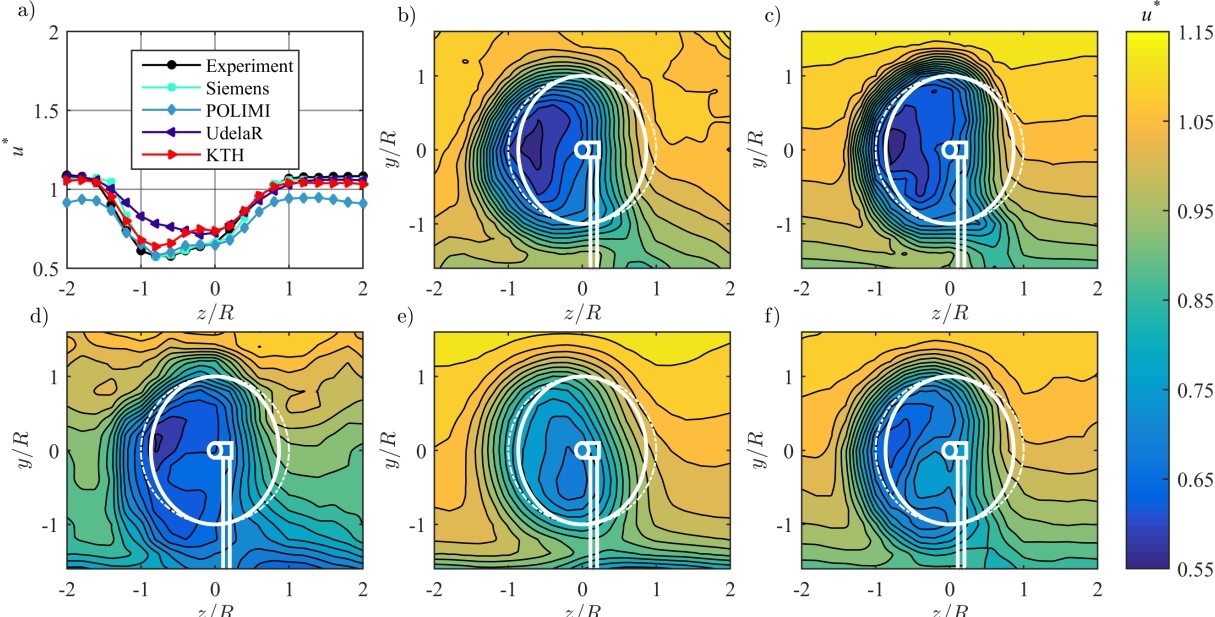

**Figure 7.** (a) Line plot and (b-f) contour plots for normalized streamwise mean velocity $u^*$ in the wake $3D$ behind turbine LARS1, from (b) experiments, (c) Siemens, (d) POLIMI, (e) UdelaR and (f) KTH. The white lines represent the turbine rotor, nacelle and tower, solid lines $\gamma = 30°$, dashed lines $\gamma = 0°$.

Next, Fig. 8 shows the normalized vertical flow component $v^*$. In general the velocity contours are dominated by two major flow patterns: A larger scale bi-pole, characterized by flow from the ceiling to the center ($v^* < 0$) and from the bottom to the center ($v^* > 0$); and a smaller bi-pole at the rotor edge at $z/D = -0.8$, where $v^*$ is positive outside the rotor swept area and negative in the rotor swept area featuring strong gradients between the peaks. These structures are generally predicted fairly well. Siemens, POLIMI and KTH match the flow pattern very accurately, which is confirmed by the line plots at hub height (Fig. 8a). High values of the correlation coefficient $r_v$ for these three simulations range from 0.819 to 0.866 and confirm the observations. The simulation by UdelaR (Fig. 8e) does not show very strong gradients and thus does not capture the detailed flow patterns. This is assumed to be due to a rather coarse mesh resolution for this simulation and can be seen in the low $r_v$-value of 0.383. Nevertheless, the general shape showing the large scale structures on the right is captured well.

The normalized turbulent kinetic energy $k^*$ is presented in Fig. 9. The contours show a clear ring of turbulence located around the rotor area. Similar to the shape of $u^*$ the ring is slightly compressed at the right side. Fig. 9a shows that all simulations predict the position and magnitude of the turbulence peaks very well. Larger differences between measurement and simulations can be found outside of the ring. Here, Siemens predicts a very low turbulent kinetic energy close to $k^* = 0$ in the free stream and in the wake center. This underprediction of $k^*$ is assumed to be due to the rather large cell size in the free stream that is too coarse to sustain the free stream turbulence. It results in a rather large $NMSE_k$ of 0.663 whereas $r_k$ with 0.873 suggests a good correlation of the shapes. POLIMI's prediction of $k^*$ shows a higher background turbulence, especially below the rotor



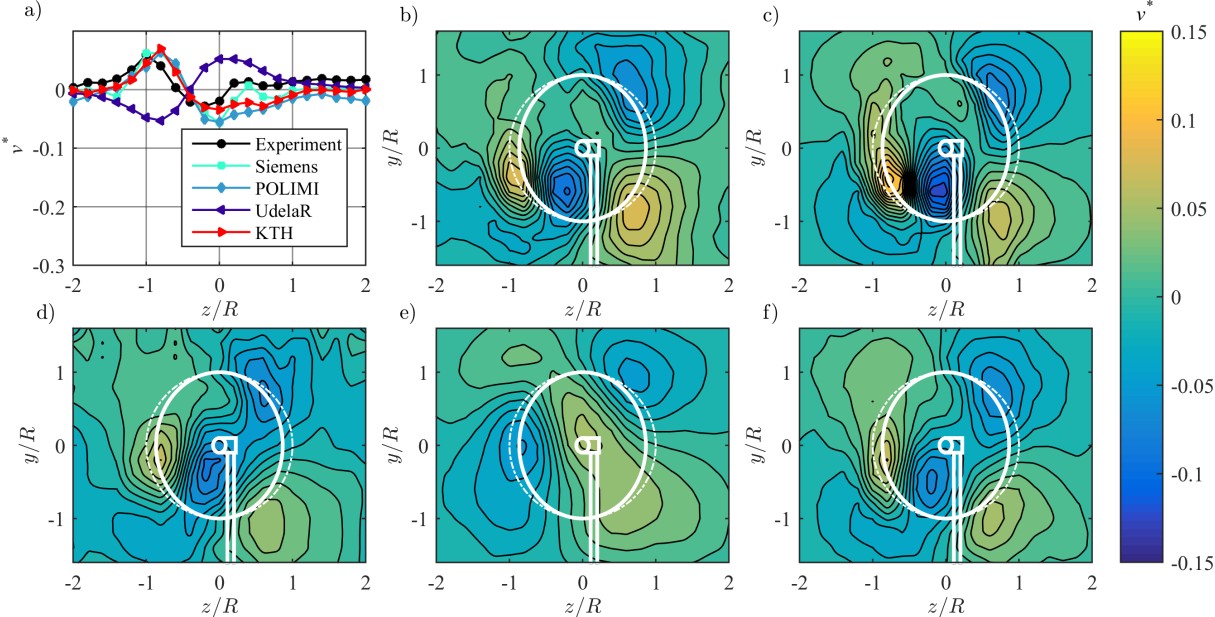

**Figure 8.** (a) Line plot and (b-f) contour plots for normalized vertical mean velocity $v^*$ in the wake $3D$ behind turbine LARS1, from (b) experiments, (c) Siemens, (d) POLIMI, (e) UdelaR and (f) KTH. The white lines represent the turbine rotor, nacelle and tower, solid lines $\gamma = 30°$, dashed lines $\gamma = 0°$.

area in positive $z$-direction. These discrepancies result in poor statistical performance values of $NMSE_k = 0.332$ and $r_k = 0.583$. UdelaR's results show a clear shear profile of $k^*$ with increasing turbulence towards the wind tunnel floor. This is quite different from the experimental results, therefore the values of $NMSE_k = 1.045$ and $r_k = 0.333$ are observed to be far off. The simulations of KTH are in very good agreement with the experiments which is confirmed by a low $NMSE_k$ of 0.085 and high $r_k$ of 0.924.

5    The comparisons of $u^*$, $v^*$ and $k^*$ $6D$ behind LARS1 show similar trends as already observed at a distance of $3D$. Therefore, the results at $6D$ are not shown. The comparison parameters summarized in Table 4 confirm these observations. A major difference to the wake at $3D$ is a more distinct curled wake shape, which is generally well predicted by all simulations. The wake is further deflected, while the skew angle is lower compared to the observations at $x = 3D$ (Table 4). This is expected to be due to the large blockage ratio of the NTNU turbine and the interference of the wake with the wind tunnel walls. The

10   experimental results of the wake at $x = 6D$ are also documented by Bartl et al. (2018a).

### 4.2   Test case 2

#### 4.2.1   Power, thrust and yaw moment

In test case 2 an aligned turbine array with both NTNU turbines LARS1 and T2 is investigated. The upstream turbine LARS1 is operated at $\gamma_{LARS1} = 30°$ and $\lambda_{LARS1} = 6.0$. Consequently, $C_{P,LARS1}$, $C_{T,LARS1}$ and $M_{y,LARS1}^*$ are identical to test case 1 (Fig. 6,





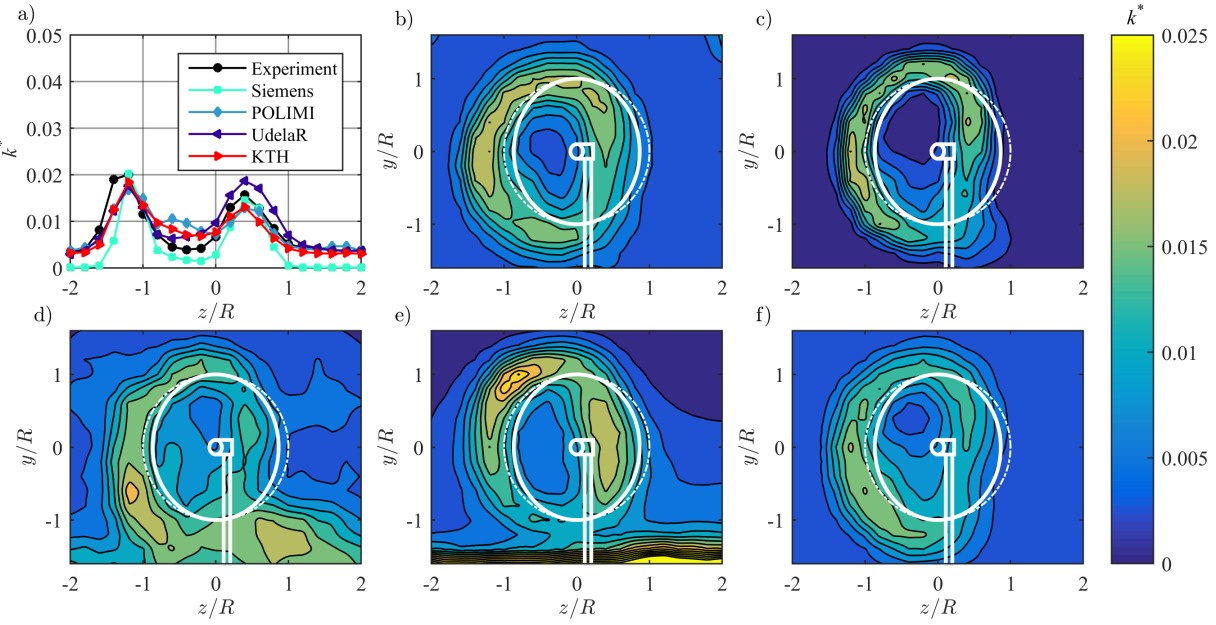

**Figure 9.** (a) Line plot and (b-f) contour plots for normalized turbulent kinetic energy $k^*$ in the wake $3D$ behind turbine LARS1, from (b) experiments, (c) Siemens, (d) POLIMI, (e) UdelaR and (f) KTH. The white lines represent the turbine rotor, nacelle and tower, solid lines $\gamma = 30°$, dashed lines $\gamma = 0°$.

**Table 4.** Comparison parameters: Skew angle ($\xi$), wake deflection ($\delta$) and available power in the wake ($P^*_{wake}$) and their differences to the measurements. Statistical performance measures: *NMSE* and $r$ for $u^*$, $v^*$ and $k^*$ at $3D$ and $6D$ behind upstream turbine LARS1.

|  | Institution | Skew angle | Deflection (z/R) | Difference (z/R) | $P^*_{wake}$ [-] | Difference [%] | $NMSE_u$ | $r_u$ | $r_v$ | $NMSE_k$ | $r_k$ |
|---|---|---|---|---|---|---|---|---|---|---|---|
| $3D$ | Experiments | 3.31° | -0.347 |  | 0.383 |  |  |  |  |  |  |
|  | Siemens | 2.53° | -0.265 | 0.082 | 0.372 | -2.7% | 0.002 | 0.964 | 0.819 | 0.663 | 0.873 |
|  | POLIMI | 3.31° | -0.347 | 0.000 | 0.340 | -11.2% | 0.017 | 0.878 | 0.830 | 0.332 | 0.583 |
|  | UdelaR | 2.92° | -0.306 | 0.041 | 0.546 | 42.7% | 0.005 | 0.914 | 0.383 | 1.045 | 0.333 |
|  | KTH | 3.31° | -0.347 | 0.000 | 0.443 | 15.6% | 0.002 | 0.957 | 0.866 | 0.085 | 0.924 |
| $6D$ | Experiments | 2.63° | -0.551 |  | 0.489 |  |  |  |  |  |  |
|  | Siemens | 2.24° | -0.469 | 0.082 | 0.476 | -2.7% | 0.002 | 0.949 | 0.810 | 0.477 | 0.898 |
|  | POLIMI | 2.44° | -0.510 | 0.041 | 0.441 | -9.9% | 0.012 | 0.860 | 0.781 | 0.164 | 0.758 |
|  | UdelaR | 2.05° | -0.429 | 0.122 | 0.691 | 41.2% | 0.006 | 0.795 | 0.463 | 0.946 | 0.192 |
|  | KTH | 2.63° | -0.551 | 0.000 | 0.527 | 7.7% | 0.002 | 0.955 | 0.805 | 0.125 | 0.970 |





Table 3) and are therefore not further discussed here. The downstream turbine T2 is operated at $\gamma_{T2} = 0°$ and the tip speed ratio $\lambda_{T2} = 5.0$ is computed using $u_{ref} = 10.0$ m/s. It is located $3D$ behind the yawed upstream turbine, meaning that the wake flow of test case 1 represents the inflow for T2. Previous Blind tests discussed the higher spread in prediction results of a downstream turbine's performance. This is confirmed by comparing $C_{P,T2}$, $C_{T,T2}$ and $M_{y,T2}^*$ of T2, which show a significantly larger spread

of performance than for test case 1 (Fig. 10, Table 3). The simulation results of the downstream turbine's power coefficient $C_{P,T2}$ (Fig. 10a) deviate between 0% and 48.9% from the experimental results. KTH matches the experimental value exactly and thus confirms the good forecast from test case 1. Siemens predicts the available power in the wake fairly accurately and thus overestimates $C_{P,T2}$ by only 10.5%. POLIMI and UdelaR over estimate $C_{P,T2}$ significantly by 43.6% and 48.9%, respectively. This trend could already be seen for the upstream turbine power coefficient $C_{P,LARS1}$ and is enhanced by overpredicting the

available power in the wake for UdelaR. POLIMI prognosticates less available power in the wake. The simulation results of the downstream turbine thrust coefficient $C_{T,T2}$ (Fig. 10b) show smaller deviations than those for $C_{P,T2}$. Nevertheless, they are slightly larger than those of $C_{T,LARS1}$ in test case 1. All simulations underestimate $C_{T,T2}$ while KTH's result shows the largest deviation of -15.3% compared to their accurate prediction of $C_{P,T2}$. Siemens and UdelaR show a similar thrust that deviates from the experimental value by -10.7% and -10.6%, respectively. POLIMI underpredicts $C_{T,T2}$ by 4.6%.

A larger spread is again observed for the simulations of $M_{y,T2}^*$ (Fig. 10c) as the values for $M_{y,T2}^*$ are very small and consequently more difficult to predict. Siemens and POLIMI are observed to overestimate $M_{y,T2}^*$ by 101.4% and 43.3%, respectively. UdelaR under predicts $M_{y,T2}^*$ for 50.6% while KTH matches the experimental results very accurately with only 1.5% difference.

### 4.2.2   Wake characteristics

This section discusses the wake characteristics $3D$ behind the two-turbine array. The wake is clearly deflected in the negative $z$-
direction. However, the deflection is not as big as $6D$ behind the single yawed turbine, but rather in the same range as $3D$ behind the single yawed turbine. This suggests that a further wake deflection is restricted by the non-yawed downstream turbine and maintained at approximately the same level at which it hits the downstream turbine. Moreover, the wake shape does not show a curled shape, instead being rather oval (Fig. 11). The tower shadow, which is mainly formed by the downstream turbine T2's tower, is more centered than in test case 1 and is well-predicted in all simulations. The shear profile in the free stream is well-
captured by all simulations. However, all predictions show a slightly lower velocity level than in the experiment. POLIMI's simulations indicate a rather strong velocity gradient again, with very low velocities close to the wind tunnel floor. However, the gradient is better established than in test case 1 as it develops further downstream. The line plot in Fig. 11a confirms that all the simulations underestimate the additional speed-up around the downstream turbine rotor. Siemens overpredicts the velocity deficit in the wake center which is confirmed by the available power that is 19.5% lower as the one resulting from
the experiments. Considering the whole wake scan, the statistical performance parameters $NMSE_u = 0.006$ and $r_u = 0.976$ on the other hand suggest better agreement. POLIMI predicts the velocities in the wake very accurately and estimates $P_{wake}^*$ only 12.1% lower than in the experiments. The statistical measures however do not confirm the good match of the energy level, resulting in a $NMSE_u$ of 0.025 and a $r_u$ of 0.925. The too low velocities in the free stream, that are not considered in $P_{wake}^*$, are deemed to impair the correlation coefficients here. The available power of UdelaR exceeds that of the experiments clearly





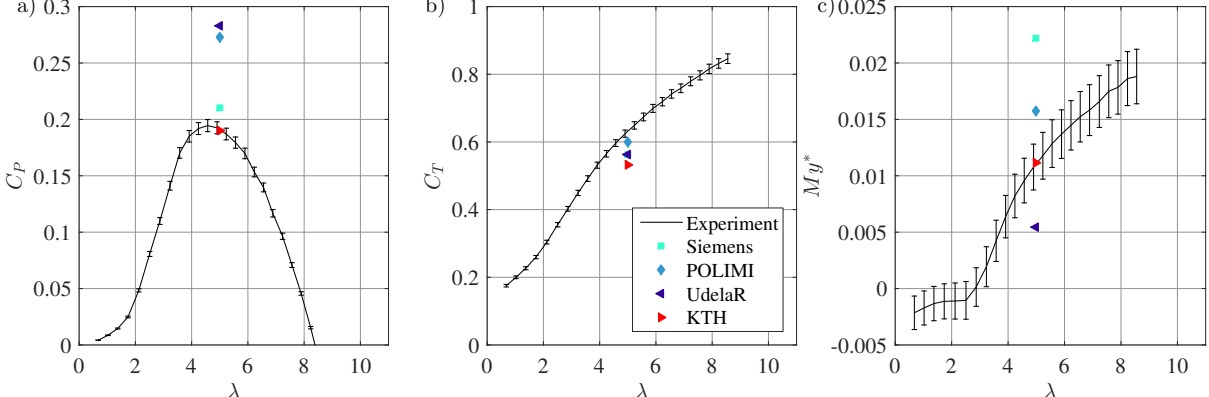

**Figure 10.** Power coefficient $C_{P,T2}$ (a), thrust coefficient $C_{T,T2}$ (b) and normalized yaw moment $M_{y,LARS1}{}^*$ (c) for the downstream turbine T2 operated in the wake of LARS1 at $\gamma_{T2} = 0°$ for $\lambda_{T2} = 1 - 12$.

by 51.1% which is mainly due to an under prediction of the velocity deficit in the wake center. Nevertheless, the statistical parameters that take the whole measurement grid into account, suggest a good agreement with $NMSE_u = 0.010$ and $r_u = 0.928$ as the lower velocities in the free stream counterbalance the higher velocities in the wake center. The velocity levels in the wake center are overpredicted by KTH, however, the available power is in good agreement with the experiments and only deviates

4.1%. This is confirmed by good statistical values of $NMSE_u = 0.007$ and $r_u = 0.976$. The wake deflection is predicted well by all simulations. POLIMI and KTH match it accurately, whereas Siemens underpredicts it by $z/R = 0.041$ and UdelaR by $z/R = 0.082$.

The contours of the vertical velocity component $v^*$ behind the turbine array show a similar flow pattern as the one behind the single yawed turbine (Fig. 12). Nevertheless, the magnitudes of $v^*$ are smaller compared to test case 1. The flow pattern is

described fairly accurately by all simulations. However, Siemens, POLIMI and KTH have average correlation values $r_v$ ranging from 0.452 to 0.586. The predictions by UdelaR are again rather coarse and thus reveal less details, which results in an even lower linear correlation coefficient of only $r_v = 0.091$.

The turbulent kinetic energy $k^*$ in the wake behind the turbine array as shown in Fig. 13 is characterized by a ring of higher TKE that is deflected in the same way as $u^*$ and thus is similar to test case 1. Compared to the single turbine wake, the ring

of high TKE is observed to be broader and flattened out (Fig. 13a). The peak locations are prognosticated very well by all simulations. However, Siemens and KTH underpredict the levels of $k^*$, while UdelaR overpredicts the turbulence in the ring, especially on the right hand side of the wake. POLIMI seems to match the turbulence in the ring fairly accurately which results in a low $NMSE_k$ of 0.087 and $r_k$ of 0.915. Good $r_k$ values are also obtained by Siemens and KTH with $r_k = 0.947$ and $r_k = 0.976$, respectively. However, their $NMSE_k$ values of $NMSE_k = 0.345$ and $NMSE_k = 0.153$, respectively, suggest some deviations. The

overprediction of TKE by UdelaR results in slightly poorer statistical performance values of $NMSE_k = 0.709$ and $r_k = 0.784$.

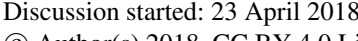



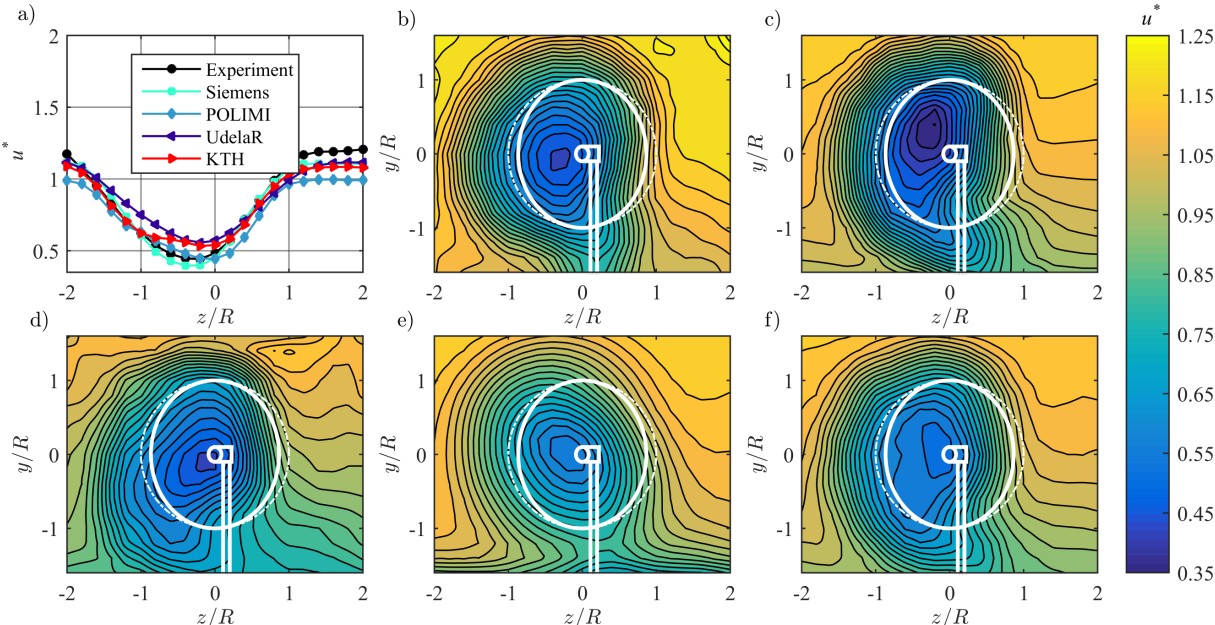

**Figure 11.** (a) Line plot and (b-f) contour plots for normalized streamwise mean velocity $u^*$ in the wake $3D$ behind downstream turbine T2, from (b) experiments, (c) Siemens, (d) POLIMI, (e) UdelaR and (f) KTH. The white lines represent the turbine rotor, nacelle and tower, solid lines $\gamma = 30°$, dashed lines $\gamma = 0°$.

**Table 5.** Comparison parameters: Skew angle ($\xi$), wake deflection ($\delta$) and available power in the wake ($P^*_{wake}$) and their differences to the measurements. Statistical performance measures: *NMSE* and *r* for $u^*$, $v^*$ and $k^*$ at $3D$ behind downstream turbine T2.

|  | Institution | Skew angle | Deflection (z/R) | Difference (z/R) | $P^*_{wake}$ [-] | Difference [%] | $NMSE_u$ | $r_u$ | $r_v$ | $NMSE_k$ | $r_k$ |
|---|---|---|---|---|---|---|---|---|---|---|---|
| 3D | Experiments | 3.71° | -0.388 |  | 0.251 |  |  |  |  |  |  |
|  | Siemens | 3.31° | -0.347 | 0.041 | 0.202 | -19.5% | 0.006 | 0.976 | 0.586 | 0.345 | 0.947 |
|  | POLIMI | 3.71° | -0.388 | 0.000 | 0.220 | -12.1% | 0.025 | 0.925 | 0.452 | 0.087 | 0.915 |
|  | UdelaR | 2.92° | -0.306 | 0.082 | 0.379 | 51.1% | 0.010 | 0.928 | 0.091 | 0.709 | 0.784 |
|  | KTH | 3.71° | -0.390 | 0.000 | 0.261 | 4.1% | 0.007 | 0.976 | 0.561 | 0.153 | 0.976 |

## 4.3 Test case 3

### 4.3.1 Wake characteristics

In the third test case the wake behind the yawed ForWind turbine is investigated. It was simulated by three of the modelers, while POLIMI did not submit predictions for this test case. The contours of the streamwise velocity $3D$ ($D = D_{ForWind}$) behind





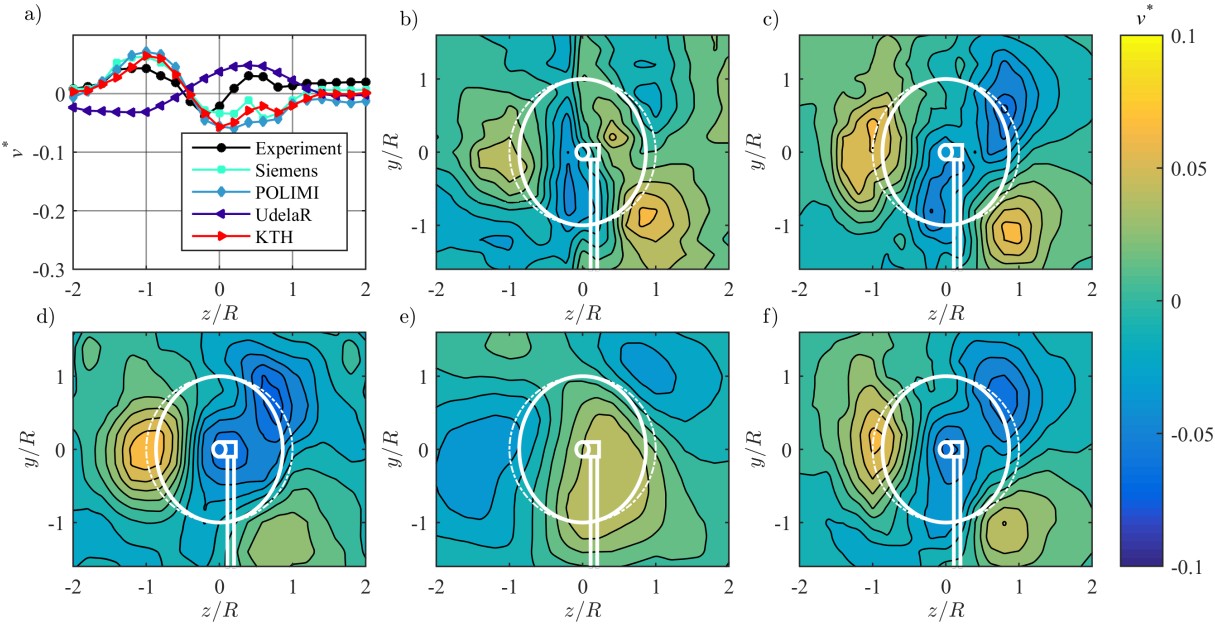

**Figure 12.** (a) Line plot and (b-f) contour plots for normalized vertical mean velocity $v^*$ in the wake $3D$ behind downstream turbine T2, from (b) experiments, (c) Siemens, (d) POLIMI, (e) UdelaR and (f) KTH. The white lines represent the turbine rotor, nacelle and tower, solid lines $\gamma = 30°$, dashed lines $\gamma = 0°$.

the ForWind turbine are presented in Fig. 14b-e. They show a more distinct curled wake shape than that observed for the NTNU turbine. In contrast to the NTNU turbine the ForWind turbine rotates in clockwise direction when observed from upstream. A counter-clockwise wake rotation deflects the wake center to the lower half behind the rotor as described in detail by Schottler et al. (2018). Furthermore, it can be seen that due to the smaller rotor diameter there is less blockage which reduces the

speed up around the rotor significantly (Fig. 14a). Thus, a smooth shear profile is observed in the free stream. The velocity deficit as well as the curled wake shape are predicted very well by all simulations with only UdelaR's simulations showing a less distinct curl. The position of the largest velocity deficit is consistent for all simulations. Nevertheless, most participants overestimate the magnitude of the velocity deficit. Siemens has the largest deviations from the experiments, which results in an available power that is 49.4% lower compared to the measurements. However, when not only taking the imaginary rotor area

into account but considering the whole wake scan, the statistical performance values $NMSE_u = 0.012$ and $r_u = 0.968$, indicate a good agreement. UdelaR predicts velocities that result in only 27.6% less available power for a potential downstream turbine, but $NMSE_u = 0.007$ and $r_u = 0.953$ are in the same range as the Siemens predictions and indicate a good match of the whole wake scan. The KTH simulation matches the experimental results best and shows the smallest deviation of available power and with $NMSE_u = 0.005$ and $r_u = 0.960$ their statistical performance values confirm the good agreement. The wake of the ForWind

turbine is slightly stronger deflected than $3D$ behind the NTNU turbine (Table 6). Siemens again under predicts the deflection, whereas UdelaR and especially KTH predict a stronger deflection of the wake than observed in the experiments.





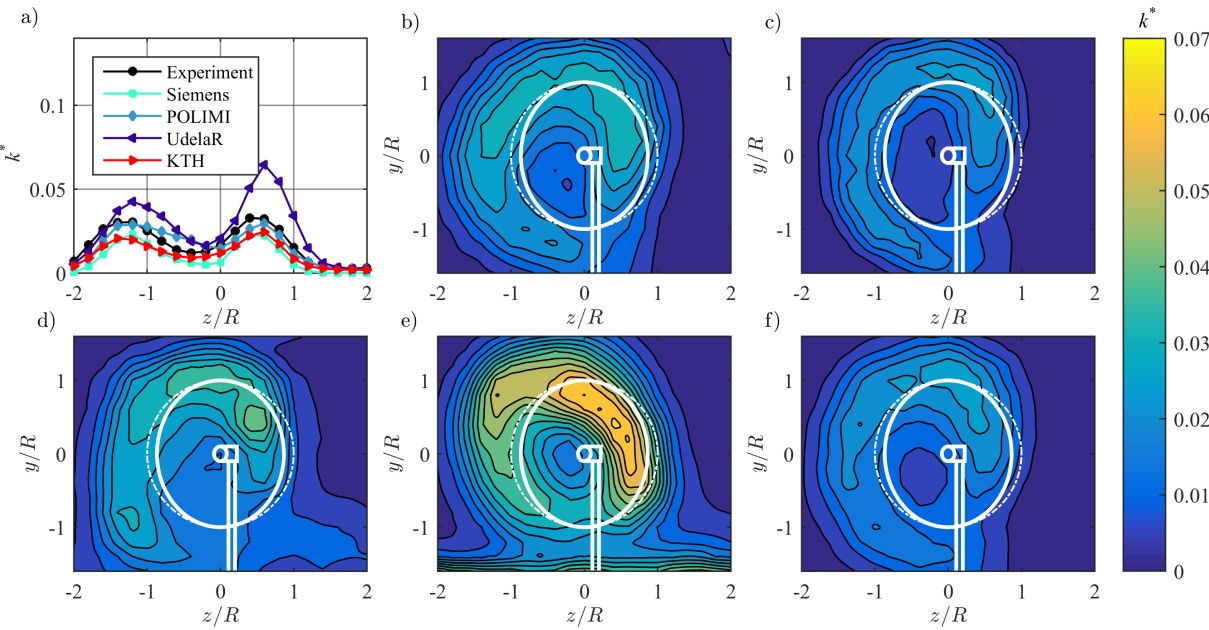

**Figure 13.** (a) Line plot and (b-f) contour plots for normalized turbulent kinetic energy $k^*$ in the wake $3D$ behind downstream turbine T2, from (b) experiments, (c) Siemens, (d) POLIMI, (e) UdelaR and (f) KTH. The white lines represent the turbine rotor, nacelle and tower, solid lines $\gamma = 30°$, dashed lines $\gamma = 0°$.

The contours of the normalized vertical velocity $v^*$ (Fig. 15b-e) are similar to those observed $3D$ behind LARS1. The flow field is dominated by the same major flow patterns as already observed in test case 1. The major difference is that the peaks in positive $z$-direction are more centered and that the dipole at the left rotor edge are not as distinct. All simulations of $v^*$ match the experiment fairly accurately , which results in similar $r_v$ values ranging from 0.802 to 0.851. Siemens however predicts

slightly higher positive peaks, but the distribution of $v^*$ is captured very well. The same applies for KTH and UdelaR, who again predicts smoother gradients due to a coarse mesh resolution.

The turbulent kinetic energy contours presented in Fig. 16b-e also indicate a clear curled shape. The $k^*$ values behind the ForWind turbine are observed to result in a significantly wider peak in positive $z$-direction (Fig. 16a) than observed behind LARS1. In contrast to the previous test cases, $k^*$ is distributed more smoothly over the wake which results in higher turbulence

levels in the wake center. The shape of the turbulent kinetic energy contours is represented accurately by all simulations. Siemens and UdelaR, however, over estimate the peak magnitudes significantly, while Siemens predicts the peak location in the upper half accurately. UdelaR's simulation is observed to result in higher TKE values in the whole ring. The simulations of KTH are in closest agreement with the experiments. The linear correlation coefficients are in the same range ($r_k = 0.878 -$ 0.905) for all three predictions. Larger deviations can be observed in $NMSE_k$ that ranges from 0.202 to 0.734.

The comparison of the wake characteristics $6D$ behind the yawed ForWind turbine results in conclusions similar to those at $3D$. Therefore, the figures comparing $u^*$, $v^*$ and $k^*$ $6D$ behind the ForWind turbine are not shown here, but the comparison





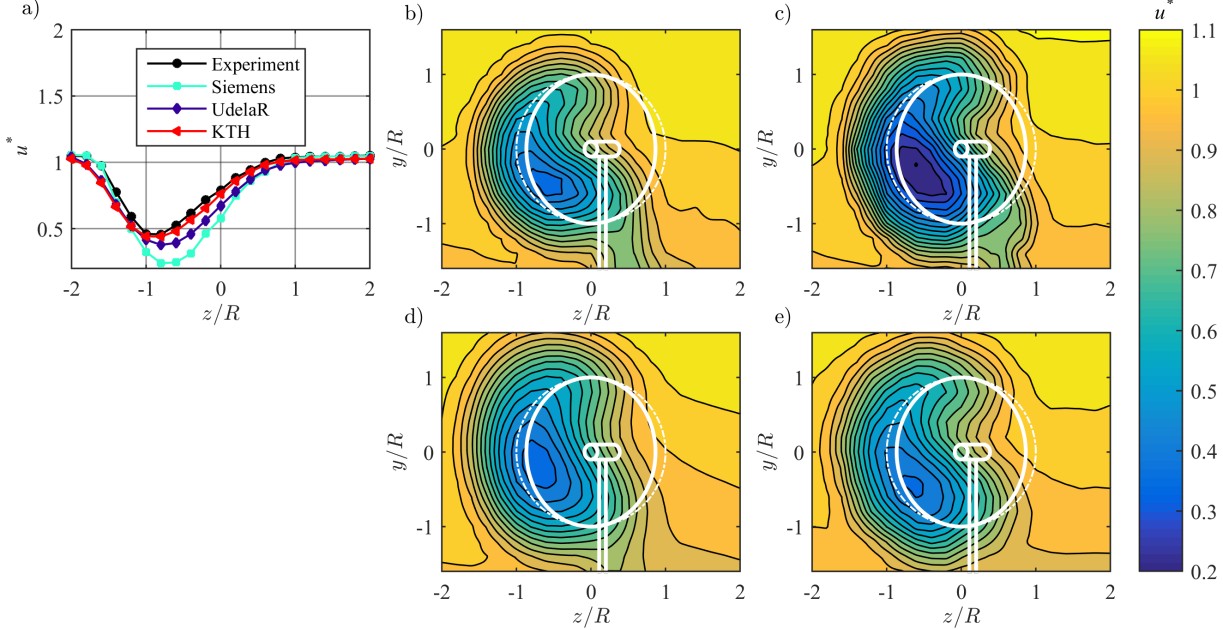

**Figure 14.** (a) Line plot and (b-e) contour plots for normalized streamwise mean velocity $u^*$ in the wake $3D$ behind ForWind turbine, from (b) experiments, (c) Siemens, (d) UdelaR and (e) KTH and. The white lines represent the turbine rotor, nacelle and tower, solid lines $\gamma = 30°$, dashed lines $\gamma = 0°$.

**Table 6.** Comparison parameters: Skew angle ($\xi$), wake deflection ($\delta$) and available power in the wake ($P^*_{wake}$) and their differences to the measurements. Statistical performance measures: *NMSE* and $r$ for $u^*$, $v^*$ and $k^*$ at $3D$ and $6D$ behind upstream ForWind turbine.

|  | Institution | Skew angle | Deflection ($z/R$) | Difference ($z/R$) | $P^*_{wake}$ [-] | Difference [%] | $NMSE_u$ | $r_u$ | $r_v$ | $NMSE_k$ | $r_k$ |
|---|---|---|---|---|---|---|---|---|---|---|---|
| $3D$ | Experiments | 4.10° | -0.429 |  | 0.285 |  |  |  |  |  |  |
|  | Siemens | 3.71° | -0.388 | 0.041 | 0.141 | -49.4% | 0.012 | 0.968 | 0.813 | 0.383 | 0.889 |
|  | UdelaR | 4.88° | -0.510 | -0.082 | 0.207 | -27.6% | 0.007 | 0.953 | 0.802 | 0.734 | 0.878 |
|  | KTH | 5.27° | -0.551 | -0.122 | 0.233 | -18.% | 0.005 | 0.960 | 0.851 | 0.202 | 0.905 |
| $6D$ | Experiments | 3.80° | -0.796 |  | 0.533 |  |  |  |  |  |  |
|  | Siemens | 3.41° | -0.714 | 0.082 | 0.430 | -19.3% | 0.002 | 0.960 | 0.845 | 0.047 | 0.961 |
|  | UdelaR | 4.00° | -0.837 | -0.041 | 0.540 | 1.2% | 0.001 | 0.963 | 0.799 | 0.067 | 0.956 |
|  | KTH | 4.19° | -0.878 | -0.082 | 0.475 | -11.0% | 0.002 | 0.950 | 0.884 | 0.052 | 0.947 |

parameters and statistical performance measures are listed in Table 6. The streamwise velocity $u^*$ and the vertical velocity $v^*$





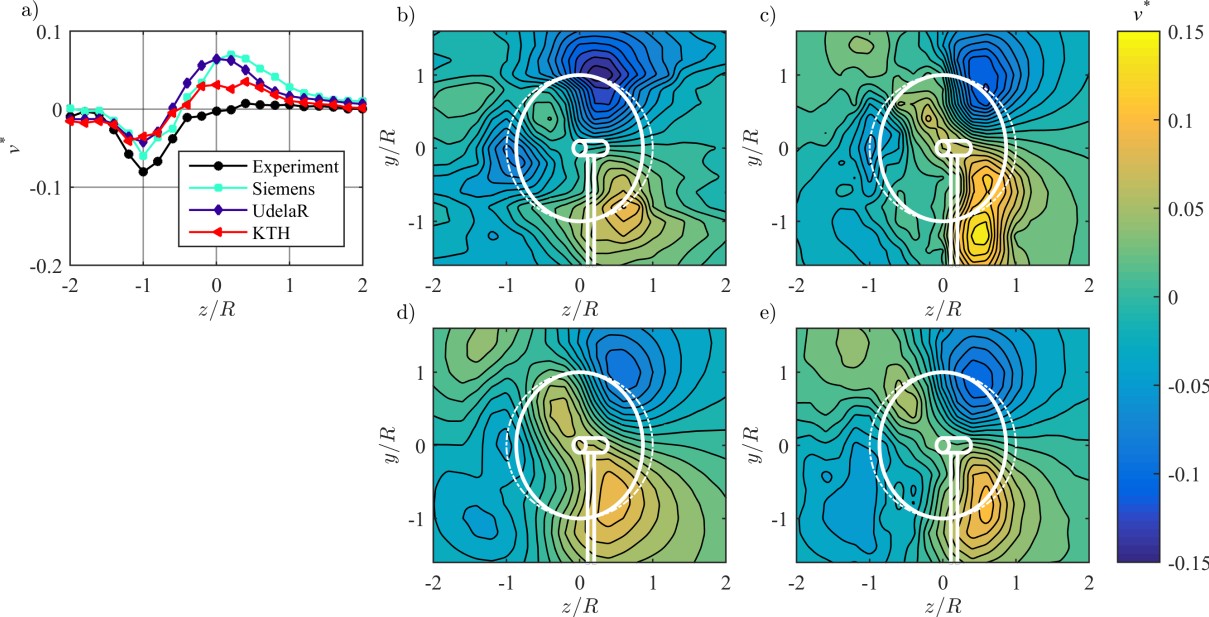

**Figure 15.** (a) Line plot and (b-e) contour plots for normalized vertical mean velocity $v^*$ in the wake $3D$ behind ForWind turbine, from (b) experiments, (c) Siemens, (d) UdelaR and (e) KTH and. The white lines represent the turbine rotor, nacelle and tower, solid lines $\gamma = 30°$, dashed lines $\gamma = 0°$.

are generally predicted accurately, which is represented by better comparison parameters and statistical performance values at $6D$ than at $3D$ for all simulations.

## 5  Discussion and conclusions

The results of four different computational contributions were compared to experimental wind tunnel results in this Blind
5  test experiment. The modelers submitted predictions for the performance of two single yawed turbine models and an aligned turbine array where only the upstream turbine is yawed. Furthermore, they predicted the mean and turbulent wake flow behind two different model turbines and the turbine array.

The power of a single yawed turbine $C_{P,LARS1}$ was predicted with a scatter of $\pm 19\%$, which was slightly bigger than in the two previous Blind test experiments. A bigger scatter of $\pm 49\%$ is observed in the predictions of the power coefficient $C_{P,T2}$
10  for a downstream turbine operating in partial wake conditions of the yawed upstream turbine. This variation is significantly larger than the scatter for an aligned downstream turbine operated in a full wake in Blind test 4 (Bartl and Sætran, 2017), in which a scatter of only $\pm 15\%$ was observed for the same distance. For a downstream turbine with a lateral offset operated in a partial wake in Blind Test 3 (Krogstad et al., 2015), however, a similar variation in power prediction was observed ($\pm 50\%$).





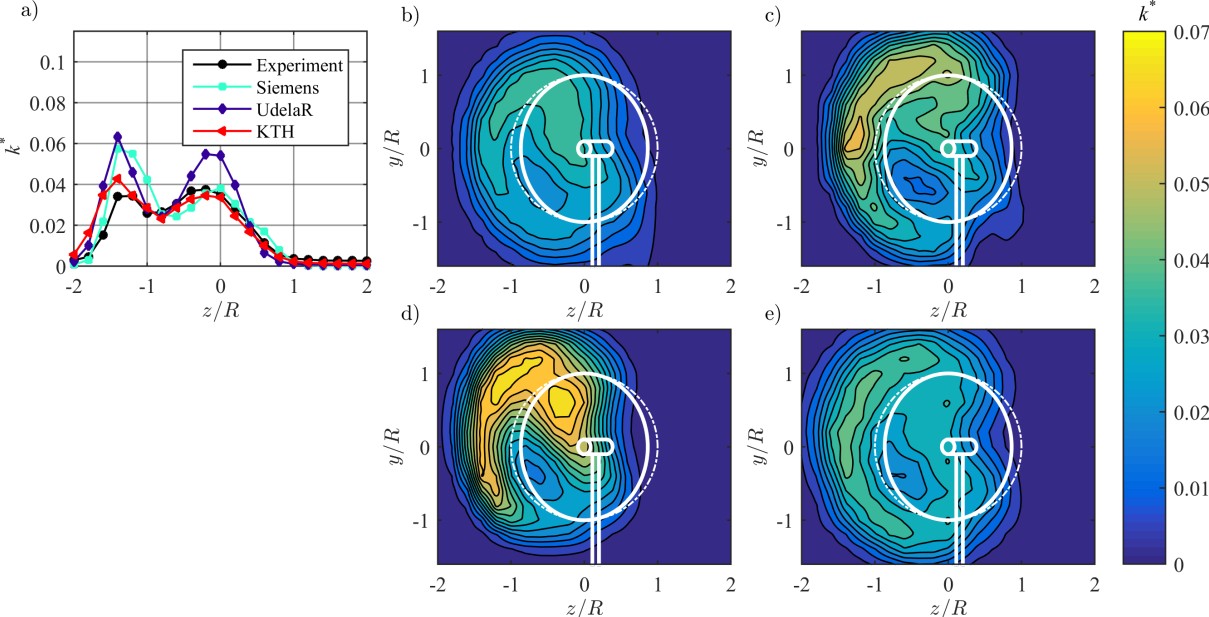

**Figure 16.** (a) Line plot and (b-e) contour plots for normalized turbulent kinetic energy $k^*$ in the wake $3D$ behind ForWind turbine, from (b) experiments, (c) Siemens, (d) UdelaR and (e) KTH and. The white lines represent the turbine rotor, nacelle and tower, solid lines $\gamma = 30°$, dashed lines $\gamma = 0°$.

These results indicate a more difficult prediction of turbine performance for an operation in a partial wake situation, due to the increased complexity of highly unsteady blade loading over the course of a rotation.

The predictions of the thrust coefficients $C_{T,LARS1}$ and $C_{T,T2}$ show a smaller scatter of $\pm 7\%$ and $\pm 15\%$, respectively, which is in the same range as observed in Blind test 4. Consequently, the thrust predictions are not influenced as strongly by yawing the turbine as the power predictions. Three of the simulations modeled the rotor by an actuator line approach, two of which used XFoil generated polars while one simulation used an experimentally measured data set. The power, thrust and yaw moment predictions of the simulations using an experimental data set consistently performed best. As the rotor was operated in yaw (test case 1) or a partial wake inflow (test case 2) the angle of attack varied during one rotor rotation, reaching high values. The experimental airfoil polars might be more realistic for such large angles of attack, which result in better performance predictions. The fourth simulation fully resolved the rotor geometry and directly calculated the forces on the rotor. The time step in these simulations was chosen to be rather large in order to save computational time which might have negatively influenced the accuracy of the blade forces. The parameters of the wake flow, however, were not impaired by this large time step.

When comparing CFD predictions to experimental measurements it is important to quantify the differences. Therefore, different techniques have been applied to analyze the wake properties. The statistical methods *NMSE* and *r* were in good agreement with each other and gave an acceptable indication of how well the simulations performed. However, they analyzed


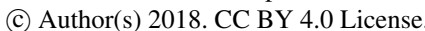


the whole wake scan and did not reveal specific discrepancies. The statistical methods were not always in accordance with the available power method, which only considered an area around the wake center for comparison. The available power method thus provided a good quantification of the wake deflection and the energy content in the wake. However, it only compared a certain section of the wake scan and accordingly could not quantify the overall performance of the simulations. Comparing

the wake contours visually resulted in a qualitative comparison, revealing flow patterns and differences in the wake shape for each simulation in comparison to the experiments . Combining the outcome from all methods provided a good overall picture of how well the wake properties from CFD predictions and measurements agree.

The comparison of the mean streamwise velocity $u^*$ in the wake generally shows a very good agreement between the experimental data and the numerical predictions. The general features such as the wake shape and deflection were predicted

well by all the simulations using IDDES as well as LES. The velocity in the wake was also predicted fairly accurately by all simulations. The high mesh resolution of the IDDES simulations by Siemens was seen to reveal exact flow details and thus resulted in a high statistical correlation for $u^*$. A similarly high statistical correlation was obtained by KTH's $u^*$ predictions using their LES-ACL simulation. The rather coarse mesh of UdelaR saved computational time, but also smeared flow details, nevertheless the velocity and turbulence levels were predicted accurately. Modeling the grid at the inlet as done in POLIMI's

simulation was observed to not perfectly predict the inflow, which was not as smooth at the position of the first turbine as in the measurements. Applying a user-defined shear profile at the inlet, as performed by the other institutions, resulted in better predictions of the free stream flow. Despite its low magnitude, the complex patterns in vertical velocity component $v^*$ were in general accurately predicted by all simulations. The details of the flow were well captured by both LES and IDDES simulations. One of the most positive results of this Blind test experiment were the very accurate predictions of the turbulent kinetic energy

in the wake behind a single turbine and the two-turbine array. The prediction of wake turbulence was seen to be difficult in previous Blind test comparisons. This workshop, however, confirms the strength of LES and IDDES simulations to accurately predict rotor generated turbulence.

Furthermore, the good results of the simulations based on a lower cell-count indicate a new trend towards CFD codes, that are able to perform accurate wake flow predictions at significantly lower computational cost. This becomes especially important

for wake predictions of full scale turbines in which the dimensions and Reynolds numbers exceed those of the experiments. Consequently, simulations with a fine grid may be very hard to realize in such a case. Nevertheless, the good performance of the coarse-grid simulations in the Blind test shows that they are a promising tool for full scale wake predictions.

Overall, the results of this Blind test comparison confirm a continuous improvement in performance and wake flow predictions from Blind test 1 to Blind test 5. LES-ACL approaches as well as the hybrid IDDES technique were confirmed to be able

to perform accurate predictions, also for complex setups featuring highly unsteady flow in yawed and partial wake operation.

*Data availability.* All presented wake data in this paper is available on https://doi.org/10.5281/zenodo.1193656.

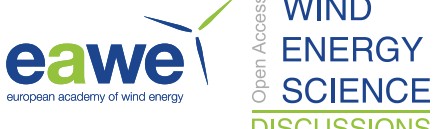

*Competing interests.* Authors declare that they have no conflict of interests

*Acknowledgements.* The authors would like to thank Stefan Ivanell and the staff of the Wind Energy group from Uppsala University, Campus Gotland for organizing the venue for the workshop.



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
