# Peer review of "Blind test comparison on the wake behind a yawed wind turbine"

_Wind Energy Science, 2018_

## Referee Comment (RC1) · Anonymous Referee #1 · 23 May 2018

Review of **Blind test comparison on the wake behind a yawed wind turbine** by *F. Muehle et al.*

GENERAL COMMENTS

This article is highly relevant to wind farm engineering, particularly in that it provides a very good overview of what state-of-the-art CFD can presently achieve in terms of prediction reliability of wind farm productivity in complex operating conditions.

SPECIFIC COMMENTS

With reference to the two bi-pole structures (contours of vertical wind speed behind the front rotor of test case 1, Fig. 8), it would be beneficial to add some discussion on the possible causes of these structures. The small-scale bi-pole structure seems to exist also behind the FprWIND turbind (test case 3, Fig. 15). It would be helpful to comment on whether such small-scale bi-pole only arises due to the wind tunnel environment, or whether, in certain conditions, it may also be observed in field operation.

With reference to the discussion on the wake characteristics of the downstream turbine (section 4.2.2.) it should be noted that the second turbine, impacted upon by the wake of the front turbine, will also generate 'its own wake', which in the absence of the oncoming wake of the upstream turbine, would not be deflected at all.  Discussion and attempts to clarify the evolution of the resulting wake (strength, direction, etc.) behind the downstream rotor, in this reviewer's opinion, ought to acknowledge the existence of the aforementioned strongly nonlinear interaction, which is indeed very relevant to the application of these results to wind farm control by means of sacrificial turbines in the front row.

TECHNICAL ORRECTIONS/COMMENTS/QUERIES

Readability could be improved by concluding Section Introduction with a clear overview of the article.

At line 15 of section 2.1 it is stated that tip Reynolds number of the NTNU turbine is 110,000. It would appear that the reference velocity used for calculating this, is the absolute wind speed of 10 m/s (this is not stated in the paper and it probably should). At line 29 it is stated that S826 was designed for Reynolds about one order of magnitude higher. However, I think that the Reynolds of 1 million refers to relative wind speed whereas that at line 15 to absolute speed. The 2 differ by a factor of 6, implying that the operational Reynolds is much closer to the design one. Please comment/amend as appropriate.

Figure 3. Since the oncoming flow is sheared, one should also indicate the orientation of the rotor angular speed because the turbine performance is in principle different depending on such sign. This information would be irrelevant only in the ideal case of zero wind shear. This information is only provided towards the end of the article, but it is suggested to add before the result section a clear schematic with the turbine, the three Cartesian axis and a graphical indication of the angular speed orientation.

Caption of Fig. 4 starts with 'Inflow at different wind tunnel positions …'. The word 'inflow', if I understand the figure correctly, may be misleading, because x/D>0 denotes positions downstream of the turbine, I assume? Please clarify/amend if required.

Section 3.3.1, line 27: please write time step as $10^{-3}$ for clarity. It would also be useful to add comments on why this value was selected, and on mesh refinement analysis to ensure reasonable independence of the computed mean results on both the spatial and the temporal resolution. These comments should be added also for the other 3 CFD set-ups. It should also be indicated what percentage of the rotor period does this number correspond to. And also if the driving criterion for this choice was to allow the development of the upstream turbulence generated by the Synthetic eddy model.

For clarity and to allow other research groups to use these results, it would be very useful to provide for each of the 4 sets of CFD simulations the distance of the inflow and outflow boundaries the distance from the first turbine along the direction of the wind stream.

Section 3.3.5, line 12. I think it's 'moment', not 'moments'.

First 2 lines of section 3.4.2 appear misplaced in that section.

Page 14, line 6: please provide clear definition of Angle of Attack in yawed wind or cite suitable reference.

Section 4.1.2. Please specify for both experimental data and numerical results whether the presented contours of streamwise velocity are averaged over a certain time interval or if they are instantaneous values. If they are averaged, please provide time interval.

Figures 7a and 8a: are these lines at hub height? Or averages along vertical direction? Please specify.

Page 19, line 1: '… and the TST … 5 is computed using $u_{ref}$=10 m/s'. This sentence is unclear.

At page 26 (Discussion and conclusions) it is stated 'The fourth simulation fully resolved the rotor geometry and directly calculated the forces on the rotor. The time-step in these simulations was chosen to be rather large in order to save computational time which might have negatively influenced the accuracy of the blade forces'. This statement presumably refers to the Siemens analyses, which used time step of 10-3 seconds. Why is this step considered small? With reference to what? Is it expected that the optimal time-step for a rotor resolved simulation should be smaller than for an ACL simulation? It would be very helpful to provide the value of the time-step for all 4 CFD simulation sets. As commented above, is the Siemens time-step too large for the synthetic eddy method although sufficient for resolving rotor unsteady aerodynamics ? Is this time-step deemed insufficient to resolve the wake turbulence? Comments on this would be very helpful to the wind farm CFD community.

The KTH simulation used measured lift and drag data. Was the maximum value of the angle of attack for which experimental data were available greater than the largest AoA expected in the 3D simulation? Or were empirical extrapolations used in the CFD look-up tables, similarly to what done in BEM analyses?

The IDDES simulations used Synthetic Eddy Method to enforce turbulent inflow fluctuations. It should be specified, however, if the other three simulation sets did something similar or used instead steady inflow conditions.

---

## Referee Comment (RC2) · G. Schepers (Referee) · 30 Jun 2018

Dear authors,

Congratulations with this excellent paper on a very thorough study. To my opinion there are far too little of these validation studies in the wind energy society!

I think the results are presented in a very clear way, although I had the feeling that the text could be a bit more concise. On the other hand I understand you want to be complete. I anyhow like the tabular summaries of these long pieces of text e.g. tables 1 and 2.

I wish I had more time to do a more thorough review but still I have a few comments:

- On some places you quantify differences between calculations and measurements but on other places you use very subjective assessments with terms like a good, poor or fair agreement. I realize very well that this is difficult to prevent but be aware that another person may come up with a completely different assessment. For example: You write on page 4, line 1 "It can be seen that the drag coefficient $C_D$ is **slightly** different". I would write that the differences are huge….
- It is good to compare CP, CT and CMyaw but please be aware that a comparison on basis of these global integrated properties has little meaning. The only conclusion you may draw from a comparison of integrated loads is that a bad agreement means that there is something wrong. A good agreement doesn't say much because we very often see 'compensating errors'. An overprediction at the root may be compensated by an underprediction at the tip or vice versa. Some examples can be found in my PhD thesis. For yawed conditions with a delicate balance between root and tip vortex effects *) I expect this to be even more the case. In that sense I think that a comparison of e.g. local yawing moments from the different calculational methods would be extremely useful to better assess the aerodynamic modelling of the different partners  (I realize this is a lot of work and it is not mandatory for me, but I strongly recommend this for a follow-up study)
- Does any of the lifting line methods, which use airfoil data, consider dynamic stall effects?
- On page 6, line 27 you write that the thrust is measured at the tower foot. I do note that the tower is included in the simulation (which is good, so you make a fair comparison) but to my point of view it still obscures the comparison a bit. Some of the differences in CT might come from the tower which are not so relevant for the wake properties since the wake is measured several ROTOR diameters (and very many TOWER diameters) behind the first turbine. In this respect: 1) Do you have "rotor off measurements", i.e. measurements of the tower alone, 2) how large is the tower drag compared to the overall thrust 3) has there been any treatment of the tower to prevent vortex shedding
- My most essential comments are about quality:
  - I am extremely happy that you include measurement uncertainties in the results Still I am a little bit surprised to see that the uncertainties are independent of the conditions. I would expect a dependency?
  - Are there any calculations which include the tunnel geometry. How large are tunnel effects? They might be more important than the measurement uncertainties in particular for yawed conditions.
  - What about the turbine quality: Have the blade geometries been scanned (small differences in blade geometry may lead to huge differences in airfoil polars at these low Reynolds numbers) and how accurate are the pitch angles? Are the pitch angles of all blades similar, and are the blade geometries similar? If not the aerodynamic

unbalance may obscure many of the results.
How accurate is the rotational speed?

I hope these comments are useful and I wish you success with the finalization of this article.

Kind regards

Gerard Schepers

*)_ For more information on this delicate balance between root and tip vortices, please have a look at:
H. Rahimi, A. Martinez Garcia, B. Stoevesandt, J. Peinke, G. Schepers (2018). *An engineering model for wind turbines under yawed conditions derived from high fidelity models*:. Wind Energy. 10.1002/we.2182.

---

## Editor Comment (EC1) · A. Bianchini (Editor) · 4 Jul 2018

Dear authors, please revise your paper according to the suggestions provided by the reviewers and post a point-to-point response to them. In particular, I would like to stress the importante of improving the paper in two specific aspects, i.e.:

1) please pay particular attention to quantify in a more effective way your results. Many times, a too qualitative discussion is provided

2) the different numerical methods need to be described in further details, in order to make the results effectively exploitable to others

Looking forward to receiving your revised paper, best regards

---

## Author Comment (AC1) · 14 Sep 2018

**Authors' response to Referee #1:**

We would like to thank the referee for reviewing this manuscript, the valuable feedback and the very constructive comments. At this stage of the review process, we respond to the referee #1's comments and propose improvements for the final manuscript. The referee's original comments are printed in **bold** followed by the corresponding answers. Passages from the manuscript are printed in *italic writing*, in which proposed additions are indicated in blue and deleted parts in .
Thank you very much for your efforts,

Franz Mühle and Jan Bartl on behalf of all authors
* * *
**Specific comment (1)**

**With reference to the two bi-pole structures (contours of vertical wind speed behind the front rotor of test case 1, Fig. 8), it would be beneficial to add some discussion on the possible causes of these structures. The small-scale bi-pole structure seems to exist also behind the ForWIND turbine (test case 3, Fig. 15). It would be helpful to comment on whether such small-scale bi-pole only arises due to the wind tunnel environment, or whether, in certain conditions, it may also be observed in field operation.**

Thank you for this very good comment. These bi-pole structures are indeed one of the most interesting features of these experiments and have recently been discussed in a number of publications by the authors (Bartl et al., 2018a; Schottler et al., 2018) and other research institutions (Howland et al, 2016; Vollmer et al., 2016; Bastankhah and Porte-Agel, 2016; Fleming et al., 2018; Berdowski et al., 2018). The discussion in the mentioned publications revealed that such structures are thought to establish for all scales, also for full-scale wind turbines in field operation.
The main purpose of the present paper is comparing the capability of different computational codes to simulate complex wake flows, without discussing the flow physics in detail. However, we suggest to add some more lines and references to the aforementioned sources in the text.

p.15, l.11 f:
*The wake contours as presented in Fig. 7b show a slightly curled wake shape. The curled wake shape was shown to develop from a counter-rotating vortex pair, as discussed in detail by Schottler et al. (2018) and Bartl et al. (2018a) for the same experimental dataset. Similar flow physics behind a yawed turbine were observed in simulations by a full scale turbine by Howland et al.(2016) and Vollmer et al. (2016). The curled wake shape is generally well predicted by three of the simulations.*

**Specific comment (2)**

**With reference to the discussion on the wake characteristics of the downstream turbine (section 4.2.2.) it should be noted that the second turbine, impacted upon by the wake of the front turbine, will also generate 'its own wake', which in the absence of the oncoming wake of the upstream turbine, would not be deflected at all. Discussion and attempts to clarify the evolution of the resulting wake (strength, direction, etc.) behind the downstream rotor, in this reviewer's opinion, ought to acknowledge the existence of the aforementioned strongly nonlinear interaction, which is indeed very relevant to the application of these results to wind farm control by means of sacrificial turbines in the front row.**

We appreciate this very good comment. We have specifically designed the second test case to be complex, i.e. the wake behind a non-yawed turbine exposed to the partial wake of a yawed upstream turbine. The results show a deflection of the wake behind the non-yawed downstream turbine as well, which is consistent with recent LES results presented by Fleming et al. (2018). The effects on power and yaw moments on the downstream turbine are presented in Bartl et al. (2018b).
Intentionally, we chose not to describe the complex flow physics of the test case in great detail. As mentioned above, our intention is to focus on comparing the capability of different computational codes to simulate complex wake flows. However, we agree with the reviewer, that a short discussion would add depth to this interesting flow phenomenon. We therefore suggest to add a couple of lines to the text:

p.18, l.9 f:
*T2 is located 3D behind the yawed upstream turbine, meaning that the wake flow of test case 1 represents the inflow for T2. Detailed results of power, thrust and yaw moments for the upstream and downstream turbine operated at different yaw angles, separation distances and inflow conditions are presented by Bartl et al. (2018b).*

p.20, l.13:
*This section discusses the wake characteristics 3D behind the two-turbine array. In this second test case the flow complexity is again increased, i.e. the wake behind a non-yawed turbine exposed to the partial wake of a yawed upstream turbine is investigated. The wake is clearly deflected in the negative z-direction. However, the deflection is not as big as 6D behind the single yawed turbine, but rather in the same range as 3D behind the single yawed turbine. This suggests that a further wake deflection is restricted by the non-yawed downstream turbine and maintained at approximately the same level, at which it hits the downstream turbine. These results compare well with a recent LES study by Fleming et al. (2018), who simulated a similar wake deflection behind a non-yawed downstream turbine exposed to a partial wake inflow.*

**Technical comment (1)**
**Readability could be improved by concluding Section Introduction with a clear overview of the article.**

We agree that it would be beneficial to include an overview. Accordingly, short summary of the article's structure is suggested to be included at the end of the Introduction section:

p.3, l.4 f:
*By increasing the complexity with respect to previous Blind tests, the wake behind a yawed wind turbine is considered to be a challenging task for simulations. The work is organized as follows. Section 2 introduces the experimental setup including a presentation of the model wind turbines and the wind tunnel and inflow conditions as well as a description of the investigated test cases. Section 3 explains the methods used in the study, including descriptions of the measurement technique, the measurement uncertainty, the applied CFD codes and the methods used for comparison. In Section 4 the experimental results and the numerical predictions for power, thrust, yaw moments and wake characteristics are presented and compared. Section 5 discusses the findings of the study, before the conclusions are stated.*

**Technical comment (2)**
**At line 15 of section 2.1 it is stated that tip Reynolds number of the NTNU turbine is 110,000. It would appear that the reference velocity used for calculating this, is the absolute wind speed of 10 m/s (this is not stated in the paper and it probably should). At line 29 it is stated that S826 was designed for Reynolds about one order of magnitude higher. However, I think that the Reynolds of 1 million refers to relative wind speed whereas that at line 15 to absolute speed. The 2 differ by a factor of 6, implying that the operational Reynolds is much closer to the design one. Please comment/amend as appropriate.**

Thank you for the comment, this might indeed be confusing and needs to be clarified. The Reynolds number of 110 000 for the NTNU turbine (line 15) and 64 000 for the ForWind turbine (line 26) were both calculated using the relative wind speed at the blade tip. The calculations are of the Reynolds numbers below.
The S826 airfoil was originally designed for higher Reynolds number of $1.0 \times 10^6$ (as stated in line 29), but used at lower Reynolds numbers in the presented experiments. For clarification, we suggest to modify the corresponding lines in the text as follows:

p.3, l.29 f:
*It is designed for Reynolds numbers of Re=$1.0\times10^6$, which is around one order of magnitude higher as the Reynolds number at the rotor tip in the presented experiments. Nevertheless, experimental data sets for airfoil performance at the lower Reynolds range around Re=$1.0 \times 10^5$ were measured  at Denmark's Technical University (DTU) (Sarlak et al., 2018) and*

*NTNU (Bartl et al., 2018c).*

$$v_{\text{Re},NTNU} = \sqrt{(10\,m/s)^2 + (60\,m/s)^2} = 60.83\,m/s$$

$$\text{Re}_{NTNU} = \frac{v_{\text{Re}} l_{chord}}{\nu} = \frac{60.83\,m/s * 0.026m}{1.46 * 10^{-5}\,m^2/s} = 108\,327$$

$$v_{\text{Re},ForWind} = \sqrt{(7.5\,m/s)^2 + (45\,m/s)^2} = 45.62\,m/s$$

$$\text{Re}_{ForWind} = \frac{v_{\text{Re}} l_{chord}}{\nu} = \frac{45.62\,m/s * 0.020m}{1.46 * 10^{-5}\,m^2/s} = 62\,493$$

**Technical comment (3)**

**Figure 3. Since the oncoming flow is sheared, one should also indicate the orientation of the rotor angular speed because the turbine performance is in principle different depending on such sign. This information would be irrelevant only in the ideal case of zero wind shear. This information is only provided towards the end of the article, but it is suggested to add before the result section a clear schematic with the turbine, the three Cartesian axis and a graphical indication of the angular speed orientation.**

Figure 3 is suggested to be complemented with the definition of the Cartesian axes and the rotational direction of the rotor. Furthermore, the rotational direction of the turbine is suggested to be added in the caption. See Figure 1 of this document.

**Technical comment (4)**

**Caption of Fig. 4 starts with 'Inflow at different wind tunnel positions ...'. The word 'inflow', if I understand the figure correctly, may be misleading, because $x/D > 0$ denotes positions downstream of the turbine, I assume? Please clarify/amend if required.**

Thank you for pointing this out, this is indeed a misleading labeling. Actually, it is the flow measured in the empty wind tunnel. It is suggested to be changed in the caption to 'Vertical flow profiles in the empty wind tunnel at different positions, in which $x/D = 0$ refers to the position, where the NTNU turbine is thereafter located'.

**Technical comment (5)**

**Section 3.3.1, line 27: please write time step as $10^{-3}$ for clarity. It would also be useful to add comments on why this value was selected, and on**

[Figure]

Figure 1: **Suggested version of Figure 3:** Sketches of the model wind turbines with reference coordinate system, (a) NTNU turbine LARS1 rotating in counterclockwise direction (CCW), (b) NTNU turbine T2 rotating in counterclockwise direction (CCW), (c) ForWind turbine rotating in clockwise direction (CW).

**mesh refinement analysis to ensure reasonable independence of the computed mean results on both the spatial and the temporal resolution. These comments should be added also for the other 3 CFD set-ups. It should also be indicated what percentage of the rotor period does this number correspond to. And also if the driving criterion for this choice was to allow the development of the upstream turbulence generated by the Synthetic eddy model.**

Thank you for this very constructive comment. These are indeed very important modeling parameters, that need to be included in the paper. In general, the modelers affirmed that their results are spatially and temporarily independent and have run corresponding sensitivity analyses.

The time step is changed to $10^{-3}$ in section 3.3.1., line 27, as suggested. Also, the descriptions of all the CFD codes are suggested to be extended by explanations on the selection of the time step and the mesh refinement analysis. Table 2 is furthermore extended, now including the 'time step [s]' and 'recording interval [s]'.

p.8, l.12 ff:

[revised manuscript text omitted]

**Technical comment (6)**

**For clarity and to allow other research groups to use these results, it would be very useful to provide for each of the 4 sets of CFD simulations the distance of the inflow and outflow boundaries the distance from the first turbine along the direction of the wind stream.**

Thank you for this comment. The distances are included in the updated descriptions of the CFD codes as presented above.

**Technical comment (7)**

**Section 3.3.5, line 12. I think it's 'moment', not 'moments'.**

We think that the plural form moments is also correct here, as it refers to several values of the requested yaw moment for the different test cases . We referred to power coefficients, thrust coefficients and yaw moments.

**Technical comment (8)**

**First 2 lines of section 3.4.2 appear misplaced in that section.**

We agree, the two lines confuse the reader and are not necessary. Therefore, we suggest to remove the first two lines of this section.

p.12, l.20 f:

**Technical comment (9)**
**Page 14, line 6: please provide clear definition of Angle of Attack in yawed wind or cite suitable reference.**

This is a very good point. We used the same definition as in the non-yawed case, where the angle of attack is defined as the angle between the relative flow direction and the chord of the blade. However, the analysis of the angle of attack was not the main focus of this study. A deeper analysis of this is given by Morote (2016).
The goal of our 2-dimensional analysis is to show the angle of attack strongly varies in the course of one rotation, and to give estimates of which flow regime the airfoils might operated in. We agree, that we should mention this simplified approach in the text and therefore suggest the following additions:

p.14, l.6 ff:
*The calculations showed that the angle of attack for the yawed turbine is fluctuating about* 2.0° *during one rotation in the outer third of the blade, causing very high angles of attack* . *Note, that the definition of the angle of attack is herein based on a simplified two-dimensional analysis, which omits the lateral component in the relative velocity during yaw.*

**Technical comment (10)**
**Section 4.1.2. Please specify for both experimental data and numerical results whether the presented contours of streamwise velocity are averaged over a certain time interval or if they are instantaneous values. If they are averaged, please provide time interval.**

All the data are time-averaged over a certain interval. For the experimental data, this is stated in section 3.3.6 and repeated in section 4.1.2. The time interval length is provided in chapter 3.1, indicating an average sampling time was approximately 25 - 33 s, depending on the non-constant data acquisition rate of the LDV system. The time interval used in the four numerical simulations is provided in the updated code descriptions (see further above in this document).

**Technical comment (11)**
**Figures 7a and 8a: are these lines at hub height? Or averages along vertical direction? Please specify.**

Thank you for pointing this out. The wake profiles are measured at hub height. The

captions of Figures 7 – 16 are therefore completed by this information: "Line plot at hub height...".

**Technical comment (12)**
**Page 19, line 1: '... and the TST ... 5 is computed using $u_{ref} = 10m/s$'. This sentence is unclear.**

Thank you for the comment. We think that it is important that we mention that the TSR of 5 for the downstream turbine was calculated using the reference velocity of 10 m/s and not the actual velocity in the wake. Nevertheless, we agree that the formulation of the sentence is unclear. Therefore, the sentence was divided into two sentences to make it clearer what we mean.

p.19, l.1 f:
*The downstream turbine T2 is operated at $\gamma_{T2} = 0°$ and $\lambda_{T2} = 5.0$. The tip speed ratio $\lambda_{T2} = 5.0$ is computed using the far-upstream reference velocity $u_{ref} = 10.0$ m/s.*

**Technical comment (13)**
**At page 26 (Discussion and conclusions) it is stated 'The fourth simulation fully resolved the rotor geometry and directly calculated the forces on the rotor. The time-step in these simulations was chosen to be rather large in order to save computational time which might have negatively influenced the accuracy of the blade forces'. This statement presumably refers to the Siemens analyses, which used time step of 10-3 seconds. Why is this step considered small? With reference to what? Is it expected that the optimal time-step for a rotor resolved simulation should be smaller than for an ACL simulation? It would be very helpful to provide the value of the time-step for all 4 CFD simulation sets. As commented above, is the Siemens time-step too large for the synthetic eddy method although sufficient for resolving rotor unsteady aerodynamics ? Is this time-step deemed insufficient to resolve the wake turbulence? Comments on this would be very helpful to the wind farm CFD community.**

Thank you for pointing this out. This is indeed an unclear and aslo incorrect description and needs revision. In Siemens' simulations, not the time step is large, but the time interval is rather short (compared to the other simulations in this BT). This short time interval might not be sufficiently long to calculate the correct time-averaged blade forces. Nevertheless, the high computational cost for the fully-resolved Siemens setup allow only short simulations, which is considered to be a major drawback of this type of simulation. The values of the time step for all simulation are included in the individual code descriptions above and Table 2. The Conclusions are suggested to be modified as follows:

p.26, l.10 f:

*The experimental airfoil polars might be more realistic for such large angles of attack, which result in better performance predictions. The  IDDES simulation fully resolved the rotor geometry and directly calculated the forces on the rotor. The  length of the simulation interval was chosen to be rather  short in order to save computational time. This might have influenced the accuracy of the time-averaged blade forces. The parameters of the wake flow, however, were not observed to be impaired by the short averaging interval.*

**Technical comment (14)**

**The KTH simulation used measured lift and drag data. Was the maximum value of the angle of attack for which experimental data were available greater than the largest AoA expected in the 3D simulation? Or were empirical extrapolations used in the CFD look-up tables, similarly to what done in BEM analyses?**

According to the tables provided by Sarlak et al. (2018), angles of attack ranging from $\alpha = -10°$ to $25°$ were available from the experimental dataset. For actuator line codes, empirical extrapolations for higher (or lower) AoAs, similar to BEM codes, are used.

**Technical comment (15)**

**The IDDES simulations used Synthetic Eddy Method to enforce turbulent inflow fluctuations. It should be specified, however, if the other three simulation sets did something similar or used instead steady inflow conditions.**

This is a very good comment, that mostly is answered in the updated descriptions of the simulation methods. Polimi actually fully resolved the turbulence and shear generating grid used in the experiment, while UdelaR and KTH used a sinusoidal modes with random phase shifts scaled with a von Karman energy spectrum, and run a precursor simulation to match the experimentally measured inflow conditions.

---

## Author Comment (AC2) · 14 Sep 2018

**Authors' response to Referee #2:**

We would like to thank Gerard Schepers for reviewing this manuscript, the valuable feedback and the very constructive comments. At this stage of the review process, we respond to the comments and propose improvements for the final manuscript. The referee's original comments are printed in **bold** followed by the corresponding answers. Passages from the manuscript are printed in *italic writing*, in which proposed additions are indicated in blue and deleted parts in .
Thank you very much for your efforts,

Franz Mühle and Jan Bartl on behalf of all authors
* * *
**Comment (1)**
**On some places you quantify differences between calculations and measurements but on other places you use very subjective assessments with terms like a good, poor or fair agreement. I realize very well that this is difficult to prevent but be aware that another person may come up with a completely different assessment. For example: You write on page 4, line 1 "It can be seen that the drag coefficient CD is *slightly* different". I would write that the differences are huge. . . .**

Thank you for pointing this out. We agree, that a qualitative comparison of results always is very subjective. For this reason, we included quantitative comparisons methods, i.e. the statistical error measured such as the normalized mean square error (NMSE) and the correlation coefficient (r). Thus, the reader can decided him/herself, whether the differences are large or small.
As for the mentioned example, we agree that the differences are rather large than small and therefore suggest the following modification:

p.4, l.1 ff:
*It can be seen that the drag coefficient $C_D$ is  very different and lift coefficient $C_L$ is diverging significantly from an angle of attack $\alpha$ of approximately 4° between the experimental and XFoil data. This difference is very distinct for high angles of attack that may occur close to stall.*

**Comment (2)**
**It is good to compare CP, CT and CMyaw but please be aware that a comparison on basis of these global integrated properties has little meaning. The only conclusion you may draw from a comparison of integrated loads is that a bad agreement means that there is something wrong. A good agreement doesn't say much because we very often see 'compensating errors'. An overprediction at the root may be compensated by an underprediction**

at the tip or vice versa. Some examples can be found in my PhD thesis. For yawed conditions with a delicate balance between root and tip vortex effects (Rahimi et al., 2018) I expect this to be even more the case. In that sense I think that a comparison of e.g. local yawing moments from the different calculational methods would be extremely useful to better assess the aerodynamic modelling of the different partners (I realize this is a lot of work and it is not mandatory for me, but I strongly recommend this for a follow-up study).

Thank you for this very good comment. It is absolutely true, that a comparison of these integral parameters CP, CT and CMyaw only allows for a very rough comparison. It is not sure at all, that the airflow over the blade is well-modeled and that different errors might compensate each other. However, we want to point out again, that the main focus of this study is comparing different modeling methods for the wake flow under complex turbine operating conditions. The CP, CT and CMyaw should rather be regarded as input variables. If these are modeled incorrectly, we expect to also see that in the wake flow.

For a more detailed comparison of the actual flow around the blades, a significantly improved experimental setup would have been necessary. This was done in the MexNext project and model turbines as the Berlin Research Turbine would allow for this kind of comparison. As our wind tunnel is limited in size, also our models are limited and consequently the equipment we can allow to be rotated in the blades.

**Comment (3)**
**Does any of the lifting line methods, which use airfoil data, consider dynamic stall effects?**

This is another very good comment. The answer is no, unfortunately not, as we consider dynamic stall effects to influence the blade aerodynamics to some degree. We consider this topic as something to look deeper into, given the simulation data available. We have updated the description of the computational methods, now including information that they are not using a dynamic stall model. (For the suggested updated descriptions see "Answers to Reviewer 1" document.)

**Comment (4)**
**On page 6, line 27 you write that the thrust is measured at the tower foot. I do note that the tower is included in the simulation (which is good, so you make a fair comparison) but to my point of view it still obscures the comparison a bit. Some of the differences in CT might come from the tower which are not so relevant for the wake properties since the wake is measured several ROTOR diameters (and very many TOWER diameters) behind the first turbine. In this respect: 1) Do you have "rotor off measurements", i.e. measurements of the tower alone, 2) how large is the tower drag compared to the overall thrust 3) has there been any treatment of the**

**tower to prevent vortex shedding?**

This is again a very good comment. 1) Yes, we have previously made "rotor-off" measurements of the tower-thrust only and 2) the tower thrust $C_{T,Tower} \approx 0.10$ at $\lambda = 6.0$, which corresponds to a little less than 10% of the total thrust. 3) On the upstream turbine LARS1, there has not been made any treatment suppressing vortex shedding. This might for practical applications be beneficial, but would add increased complexity to the test cases.
In contrast to earlier Blind test experiments, where we have tried to compare the rotor thrust only, we now decided to include the tower in the comparison. Earlier, we had to subtract the tower thrust from the measured total thrust, which itself included some uncertainties, as the major part of the tower is in the wake of the rotor. This solution seemed to be the fairest, for comparison purposes.

**Comment (5)**
**I am extremely happy that you include measurement uncertainties in the results. Still I am a little bit surprised to see that the uncertainties are independent of the conditions. I would expect a dependency?**

The measurement uncertainties are indeed different for the different test cases and operating conditions. This can be observed in Figures 6 and 10, where CP, CT and CMyaw are compared. The uncertainty values presented in Section 3.2 indicate the maximum calculated uncertainties measured for the different variables.

**Comment (6)**
**Are there any calculations which include the tunnel geometry. How large are tunnel effects? They might be more important than the measurement uncertainties in particular for yawed conditions.**

This is a very good thought. Indeed, all numerical simulations include the wind tunnel geometry as it certainly affects the performance and wake development (specifically wake deflection in yawed conditions). It would have been very interesting to quantify the effects of tunnel blockage on turbine performance and wake development, by running a simulation with and without the wind tunnel boundaries. This is, however, considered to be a specific study on its own. A study of the effects of different wind tunnel blockage ratios on the wake development and expansion of a non-yawed turbine has previously been performed by Sarlak et al. (2016), using the geometry of our NTNU model turbine. A follow up study on the effects of yaw would be very interesting.

**Comment (7)**
**What about the turbine quality: Have the blade geometries been scanned**

(small differences in blade geometry may lead to huge differences in airfoil polars at these low Reynolds numbers) and how accurate are the pitch angles? Are the pitch angles of all blades similar, and are the blade geometries similar? If not the aerodynamic unbalance may obscure many of the results. How accurate is the rotational speed?

Thank you for this very good comment focusing on the accuracy of the experiment. No, the blade geometries have not been scanned by a 3D scanner. We agree, that small geometry inaccuracies (which are definitely present) might influence the wake results more than expected. We have, however, used optical laser methods as well as phase-locked photographic methods to ensure that the pitch settings of our three blades are deviating as little as possible (as much as these methods allow). Furthermore, the pitch settings have been adjusted such that the vortex shedding of the three blades in the very near wake is equidistant. The distance between the three vortices shed were measured by the means of phase average hot wire measurements (Eriksen and Krogstad, 2017).

The rotational speed is constantly measured by an optical sensor in the nacelle. The variations in rpm are observed to be below 1.0%.

**References**

[1] Rahimi, H., Martinez Garcia, A., Stoevesandt, B., Peinke, J., and Schepers, G.: An engineering model for wind turbines under yawed conditions derived from high fidelity models:. Wind Energy, doi: 10.1002/we.2182., 2018.

[2] Sarlak, H., Nishino, T., Martinez-Tossas, L.A., Meneveau, C., and Sørensen, J.N.: Assessment of blockage effects on the wake characteristics and power of wind turbines, Renewable Energy, 93, 340–352, doi: 10.1016/j.renene.2016.01.101, 2016.

[3] Eriksen, P.E. and Krogstad, P.-Å: Development of coherent motion in the wake of a model wind turbine, Renewable Energy, 108, 449–460, doi: 10.1016/j.renene.2017.02.031, 2017.

---

## Referee Report (RR1)

Thanks for adressing my comments. The response is very satisfactory.

Good luck with finishing the paper

Gerard